# Cul3 regulates cytoskeleton protein homeostasis and cell migration during a critical window of brain development

Jasmin Morandell[1,3], Lena A. Schwarz[1,3], Bernadette Basilico [1], Saren Tasciyan[1], Georgi Dimchev[1], Armel Nicolas[1], Christoph Sommer [1], Caroline Kreuzinger[1], Christoph P. Dotter [1], Lisa S. Knaus[1], Zoe Dobler[1], Emanuele Cacci[2], Florian K. M. Schur [1], Johann G. Danzl[1] & Gaia Novarino [1]✉

De novo loss of function mutations in the ubiquitin ligase-encoding gene *Cullin3* (*CUL3*) lead to autism spectrum disorder (ASD). In mouse, constitutive *Cul3* haploinsufficiency leads to motor coordination deficits as well as ASD-relevant social and cognitive impairments. However, induction of *Cul3* haploinsufficiency later in life does not lead to ASD-relevant behaviors, pointing to an important role of *Cul3* during a critical developmental window. Here we show that *Cul3* is essential to regulate neuronal migration and, therefore, constitutive *Cul3* heterozygous mutant mice display cortical lamination abnormalities. At the molecular level, we found that Cul3 controls neuronal migration by tightly regulating the amount of Plastin3 (Pls3), a previously unrecognized player of neural migration. Furthermore, we found that Pls3 cell-autonomously regulates cell migration by regulating actin cytoskeleton organization, and its levels are inversely proportional to neural migration speed. Finally, we provide evidence that cellular phenotypes associated with autism-linked gene haploinsufficiency can be rescued by transcriptional activation of the intact allele in vitro, offering a proof of concept for a potential therapeutic approach for ASDs.

[1] Institute of Science and Technology (IST) Austria, Klosterneuburg, Austria. [2] Department of Biology and Biotechnology "Charles Darwin", Sapienza, University of Rome, Rome, Italy. [3] These authors contributed equally: Jasmin Morandell, Lena A. Schwarz. ✉email: gnovarino@ist.ac.at

The past decade has seen a major effort to elucidate the genetic underpinnings of autism spectrum disorders (ASDs). Whole exome sequencing of large patient cohorts and their unaffected family members has identified hundreds of ASD-risk loci[1–5]. However, the molecular and cellular functions of the majority of the identified genes remain poorly understood. One of the identified high-risk ASD genes encodes the E3 ubiquitin ligase Cullin3 (Cul3)[1–3,6–11].

E3 ubiquitin ligases regulate cellular protein composition by providing target recognition and specificity to the ubiquitin-dependent proteasomal degradation pathway[12]. CUL3 is a conserved protein of the Cullin family, comprising eight members, which contain a conserved cullin homology domain, named after its ability to select cellular proteins for degradation. CUL3 ASD-associated genetic variants are most often de novo missense or loss of function (loF) mutations, dispersed throughout the entire gene and affecting distinct protein domains. In addition to the ASD core symptoms, patients with CUL3 de novo loF mutations can present with several comorbidities including varying levels of intellectual disability (ID), attention deficit hyperactivity disorder (ADHD), sleep disturbances, motor deficits, epileptic seizures, and facial dysmorphisms[10,11,13,14]. The only known exception is the deletion of CUL3 exon 9 by a specific dominant splice site variant causing a severe form of pseudohypoaldosteronism type II (PHAII), featuring hypertension, hyperkalemia, and metabolic acidosis but not ASD[15–17]. Despite the well-understood process of CUL3-mediated protein ubiquitination and degradation[12], its target proteins in the developing central nervous system and its role in brain development remain utterly understudied.

Here, we show that Cul3 is required during brain development to regulate neuronal migration and thus precisely assemble the cerebral cortex. At the molecular level, Cul3 regulates cytoskeletal and adhesion protein abundance in mouse embryos. In particular, we found that Cul3 controls the abundance of Plastin 3 (Pls3), a novel player of neural cell migration, whose amount is inversely proportional to neural cell migration speed. Finally, we found that CRISPR-mediated activation of Cul3 transcription fully rescues neural cell migration defects. Altogether, our analysis highlights a pivotal role for Cul3 in brain development, identifies a new player of neuronal migration, and provides a proof of concept of CRISPR-mediated rescue of an ASD-linked genetic defect.

## Results

**Behavioral defects in Cul3 haploinsufficient animals.** To model ASD-linked mutations, we studied a constitutive heterozygous Cul3 knockout (Cul3[+/−]) mouse[18] (Supplementary Fig. 1a). As predicted, Cul3[+/−] animals show a significant decrease in Cul3 protein in the brain, to approximately 50% of wild-type levels (Supplementary Fig. 1b and Supplementary Data 1). Importantly, Cul3 protein reduction is equal in all brain areas tested (Supplementary Fig. 1b), thus resembling patients with germline mutations.

Although Cul3 haploinsufficient mice have a slightly reduced body weight at birth, their weight is comparable to control animals as adults (Supplementary Fig. 1c), while the brain to body weight ratio is unaffected in mutant newborn and adult mice (Supplementary Fig. 1d). Adult Cul3[+/−] mice present with hind limb clasping (Fig. 1a) and mild gait abnormalities, such as increased sway and stance length (Fig. 1b, c and Supplementary Fig. 2a), phenotypes which are observed in other ASD mouse models[19,20] and indicative of cerebellar dysfunctions[21]. Further indicating motor defects, Cul3[+/−] mice underperform when challenged on the accelerating RotaRod (Fig. 1d, d′), a task requiring formation and consolidation of a repetitive motor routine[22,23]. Mutant mice do not reach the same level of motor performance as their healthy counterparts by the end of the third day of trials, suggesting abnormal motor learning dynamics (Fig. 1d). Additionally, while no general sex differences were observed in Cul3 heterozygous knockout mice, male Cul3 haploinsufficient animals show reduced initial coordination compared to their wild-type littermates (Supplementary Fig. 2b). Motor defects of Cul3[+/−] mice, however, do not affect exploratory behavior in the open field (Supplementary Fig. 2c), nor on the elevated plus maze, where Cul3[+/−] animals do not show differences in anxiety-like behaviors (Supplementary Fig. 2d).

Next, we subjected Cul3[+/−] animals to classical sociability tests. In the three-chamber test, similarly to wild-types, Cul3[+/−] mice show a preference for a mouse (M1) over an object (Ob.) (Fig. 1e). However, in the second phase of the assay, mutant mice show no preference for a stranger mouse (M2) over a familiar animal (M1), a preference displayed by control animals (Fig. 1f). We thus concluded that haploinsufficiency of the Cul3 gene is associated with reduced interest in social novelty. As social recognition is mainly achieved via olfaction in rodents[24,25], we assessed the ability of mutant animals to distinguish and familiarize themselves with non-social and social odors. In the odor discrimination and habituation test (ODHD)[26], both wild-type and Cul3[+/−] animals successfully recognize newly and already presented odors (Supplementary Data 2). However, mutant mice spend significantly more time exploring odor-embedded cotton swabs and are hyper-reactive to the presentation of social odors (Fig. 1g and Supplementary Data 1, 2). Thus, despite mutant animals spending significantly more time sniffing social odors than controls, Cul3[+/−] mice are able to distinguish between two different social odors, indicating that the reduced social novelty interest is not directly related to odor discrimination issues. Finally, we employed a well-established memory test[27] to assess how Cul3 haploinsufficiency affects learning. Contextual fear conditioning revealed normal fear acquisition and memory retention in Cul3[+/−] mice. However, mutant animals exhibit a reduced ability to extinguish the aversive memory after extinction training, pointing towards abnormal cognition (Fig. 1h, h′).

In summary, our analysis indicates that Cul3 haploinsufficiency leads to abnormalities in several behavioral paradigms, potentially associated with dysfunction of different brain areas and/or dysfunctional brain connectivity.

**Behavioral abnormalities are associated with Cul3 developmental functions.** The point(s) in time when ASD mutations exert their effects on the brain remain elusive in most cases. However, identifying these critical temporal windows may be essential to properly design therapeutic strategies and clinical trials. In order to understand whether Cul3 haploinsufficiency is critical for the appearance of ASD-associated behaviors at developmental stages or throughout life, we analyzed the effects of Cul3 deletion at a later time point. To induce deletion of Cul3 postnatally, we crossed our conditional Cul3 allele with animals expressing a tamoxifen-responsive Cre recombinase (Cag-CreER). Thus, we induced heterozygous Cul3 deletion by tamoxifen (TM) injections of Cul3[+/fl] Cag-CreER mice between P30 and P40, and performed a behavioral analysis of these animals at P55-60. Cul3[+/fl] Cag-CreER mice injected with vehicle (V) and Cul3[+/fl] mice injected with tamoxifen were used as controls for the CreER or the compound, respectively (Fig. 2a).

Consecutive daily tamoxifen injections (100 mg/kg, 5 days, starting at P30) significantly decrease Cul3 protein to about half of control levels in Cul3[+/fl] Cag-CreER brain tissue (Fig. 2b, c). Of note, we observed that tamoxifen mediated homozygous deletion of Cul3 induced premature lethality in Cul3[fl/fl] Cag-CreER animals,

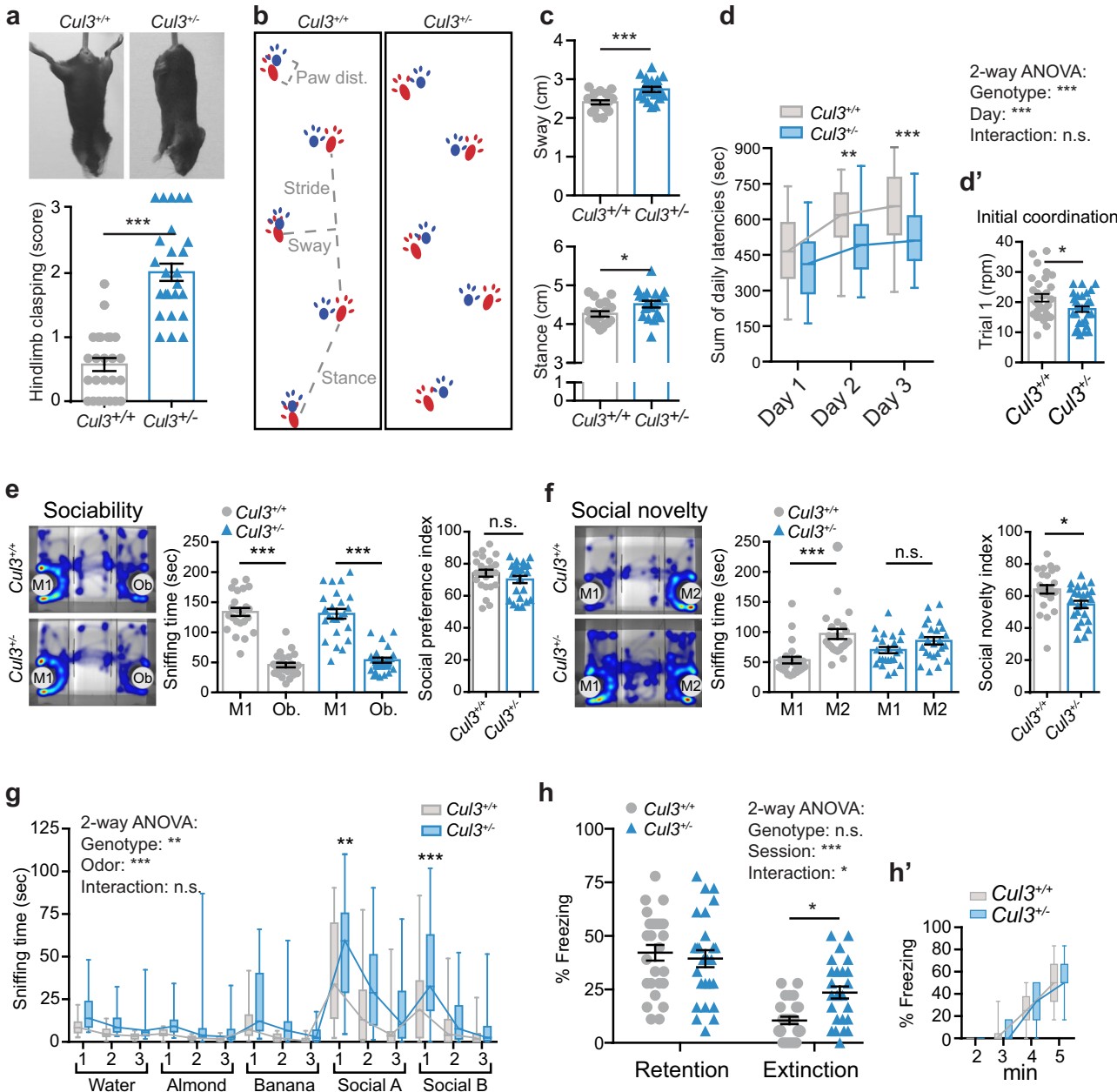

**Fig. 1 Behavioral defects in *Cul3* haploinsufficient mice. a** Hind limb clasping in adult *Cul3*[+/−] mice, not observed in wild-type littermate controls (**a** top); scoring 0–1 (normal) to 3 (most severe) (**a** bottom, $n = 25$ animals per genotype; ***$P < 0.0001$; two-tailed Mann–Whitney $U$-test). **b** *Cul3*[+/+] and *Cul3*[+/−] strides, forepaws (blue), and hind paws (red). **c** Altered gait of *Cul3*[+/−] mice evidenced by comparison of sway (**c** top) and stance length (**c** bottom) ($n = 19$ mice per genotype; *$P = 0.03$, ***$P = 0.0004$; two-tailed t-test). **d, d′** Accelerating RotaRod test revealing defects in motor performance and coordination in *Cul3*[+/−] mice. Shown: sum of daily latencies of three trials per day on three consecutive days and final rpm on day one—trial 1, i.e.: initial coordination (**d′**) ($n = 30$ mice per genotype; *$P = 0.021$, **$P = 0.001$, ***$P = 0.0005$; 2-way ANOVA and Sidak's multiple comparison test and unpaired two-tailed $t$-test). **e, f** Heat maps of three-chamber social interaction test (left), quantification of interaction times (middle), and social preference/novelty index (right). Sociability: *Cul3*[+/−] and control mice spend more time with a stranger mouse (M1) than an object (Ob.) (**e**); social novelty: *Cul3*[+/−] mice do not prefer a novel stranger (M2) over the familiar mouse (M1) (**f**) ($n = 24$ mice per genotype; *$P = 0.01$, ***$P < 0.0001$, n.s. not significant; 1-way ANOVA and Sidak's multiple comparison test). **g** Both genotypes distinguish and familiarize to non-social and social odors in the olfaction habituation and dishabituation test, yet *Cul3*[+/−] mutant mice are hyper-reactive to the presentation of social odors ($n = 24$ mice per genotype; **$P = 0.002$, ***$P = 0.0009$, n.s. not significant; 2-way ANOVA and Sidak's multiple comparison test; details in Supplementary Data 1, 2). **h, h′** Contextual memory retention and extinction scored as percent freezing during a 3 min exposure to the context (**h**), and fear-acquisition training (**h′**) ($n = 26$ mice per genotype; *$P = 0.027$, n.s., not significant; 2-way ANOVA interaction: $(F1,100) = 6.18$; $P = 0.015$; Sidak's multiple comparisons test: Extinction $P = 0.027$). Sex-matched littermate animals were analyzed. Data presented either as mean ± SEM, as well as scatter plot (**a**, **c**, **d′**, **e**, **f**, **h**) or as boxplot showing median value and 25–75th percentile, whiskers show minimum and maximum (**d**, **g**, **h′**). Detailed statistics are provided in Supplementary Data 1.

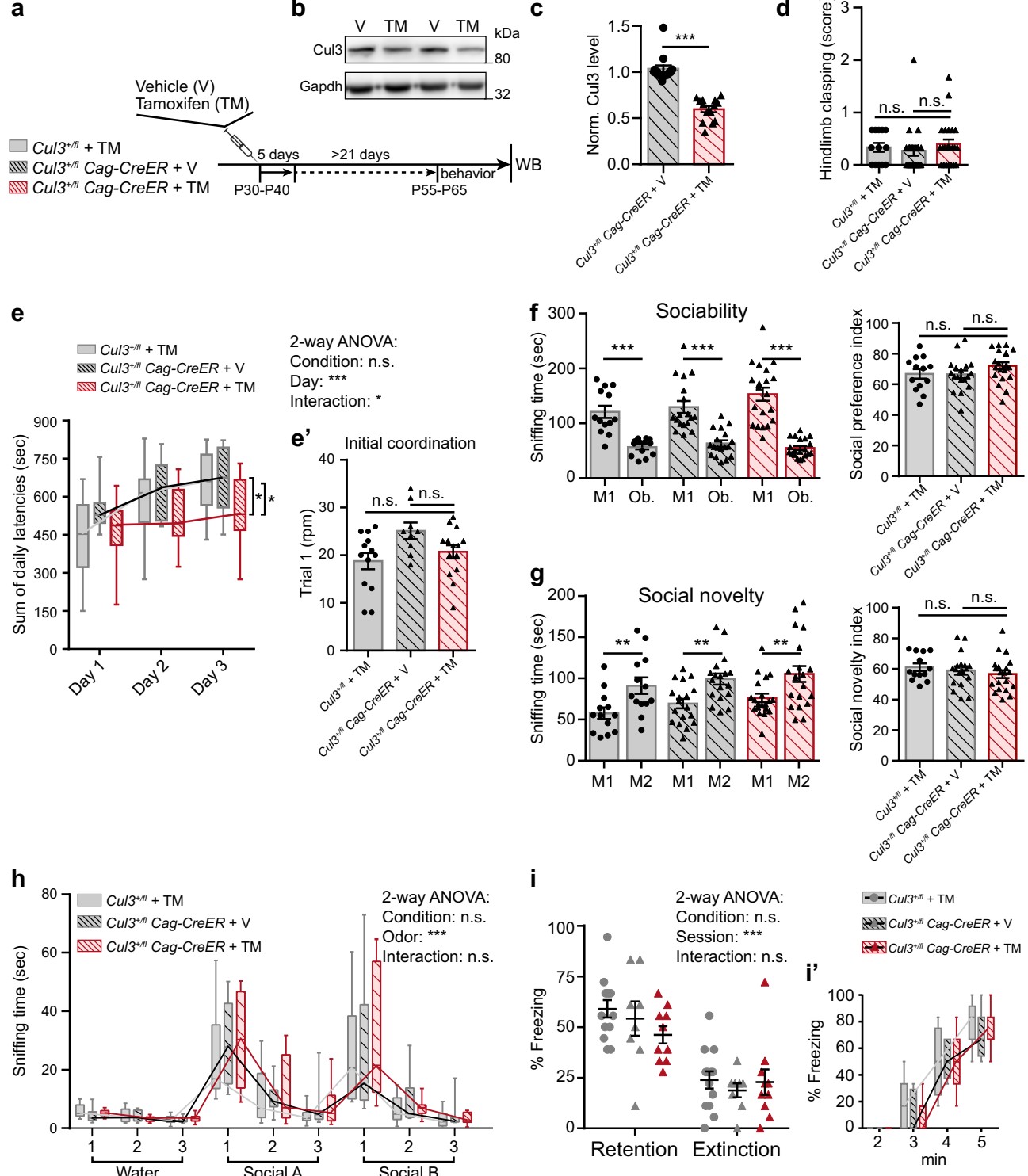

while *Cul3+/fl Cag-CreER* mice show normal survival into adulthood (Supplementary Fig. 2e, f). This finding underscores a dose-dependent effect of Cul3 depletion and highlights the necessity of employing construct valid mouse models when investigating human pathophysiology.

Next, we performed a thorough behavioral characterization of the *Cul3+/fl Cag-CreER* animals treated with tamoxifen, testing all the paradigms in which we observed behavioral abnormalities in the constitutive adult *Cul3+/−* line (Fig. 1). We found that induction of *Cul3* haploinsufficiency at P30 does not result in any

major behavioral defect, except for reduced motor performance on the accelerating RotaRod. More specifically, while we did not observe any increase in hind limb clasping events (Fig. 2d), abnormalities in gait (Supplementary Fig. 2g) or initial coordination on the RotaRod, tamoxifen injected *Cul3+/fl Cag-CreER* mice significantly underperform on the third day of RotaRod training (Supplementary Fig. 2h and Fig. 2e, e′), indicating that Cul3 function is required for motor learning in adult animals. In contrast, *Cul3+/fl Cag-CreER* treated with tamoxifen behave comparably to the controls in the three-chamber sociability test,

**Fig. 2 Cul3 loss after completion of main developmental milestones does not lead to major behavioral abnormalities in mice. a, b** Juvenile $Cul3^{+/fl}$ Cag-CreER mice were injected for 5 days with either 100 mg/kg tamoxifen (TM) or vehicle (V), $Cul3^{+/fl}$ animals were injected with TM. Behavioral tests were performed after ≥21 days post-last injection, followed by western blot analysis of brain Cul3 levels (**a** scheme; **b** representative western blot). **c** Quantification of Cul3 levels of tamoxifen-treated mice, normalized to vehicle-injected controls ($n = 14$ mice per condition; ***$P < 0.0001$; unpaired two-tailed $t$-test). **d** Hind limb clasping scoring from 0–1 (normal) to 3 (most severe) in all conditions ($n(Cul3^{+/fl} + TM) = 13$, $n(Cul3^{+/fl}$ Cag-CreER + V) = 18, $n(Cul3^{+/fl}$ Cag-CreER + TM) = 20; 1-way ANOVA and Sidak's multiple comparison test). **e, e'** Accelerating RotaRod revealed reduced motor learning abilities in $Cul3^{+/fl}$ Cag-CreER + TM mice (**e**), initial coordination was unaffected ($n(Cul3^{+/fl} + TM) = 13$, $n(Cul3^{+/fl}$ Cag-CreER + V) = 8, $n(Cul3^{+/fl}$ Cag-CreER + TM) = 14; *$P = 0.03$; 2-way ANOVA (**e**), 1-way ANOVA (**e'**) and Sidak's multiple comparison tests). **f, g** Normal behavior in three-chamber sociability test upon juvenile Cul3 loss, interaction times (left), social preference/novelty index (right). Test and control mice significantly prefer a stranger mouse (M1) over the object (Ob.) (**f**), and the novel stranger (M2) over the familiar mouse (M1) (**g**) ($n(Cul3^{+/fl} + TM) = 13$, $n(Cul3^{+/fl}$ Cag-CreER + V) = 18, $n(Cul3^{+/fl}$ Cag-CreER + TM) = 20; *$P < 0.05$, ***$P < 0.0001$; 1-way ANOVA and Sidak's multiple comparison test). **h** All test-groups distinguish and familiarize similarly to social odors in the adapted olfaction habituation and dishabituation test ($n(Cul3^{+/fl} + TM) = 13$, $n(Cul3^{+/fl}$ Cag-CreER + V) = 8, $n(Cul3^{+/fl}$ Cag-CreER + TM) = 10; 2-way ANOVA and Sidak's multiple comparison test; details: Supplementary Data 1, 2). **i, i'** Contextual memory retention and extinction scored as percent freezing during exposure to context (**i**), and fear-acquisition training (**i'**) ($n(Cul3^{+/fl} + TM) = 13$, $n(Cul3^{+/fl}$ Cag-CreER + V) = 8, $n(Cul3^{+/fl}$ Cag-CreER + TM) = 10; 2-way ANOVA and Sidak's multiple comparison test). Sex-matched littermates were analyzed. Data presented either as mean ± SEM, as scatter plot (**c, d, e', f, g, i**) or as boxplot showing median value and 25–75th percentile, whiskers show minimum and maximum (**e, h, i'**). Significance levels: *$P < 0.05$, **$P < 0.01$, ***$P < 0.001$, n.s. not significant. Detailed statistics are provided in Supplementary Data 1.

with normal interest in social novelty (Fig. 2f, g). Further, they react normally to the social odors in an adapted OHDH test (Fig. 2h and Supplementary Data 2) and show comparable cognitive abilities in the CFC paradigm to the controls (Fig. 2i, i'). All of these behaviors were clearly perturbed in animals with germline Cul3 haploinsufficiency but appear normal when Cul3 is deleted at the juvenile stage. These results indicate that developmental stages are critical for the appearance of Cul3-associated behavioral phenotypes.

**Cul3 haploinsufficiency is associated with abnormal brain development.** Having established that ASD-relevant behaviors are associated with Cul3 functions in the developing brain, we investigated its expression pattern. Temporally, in the mouse brain, Cul3 expression peaks at E14.5 and E16.5 (Supplementary Fig. 3a, b). Spatially, it is predominantly expressed in the cortex and hippocampus (Supplementary Fig. 3c), in glutamatergic and inhibitory neurons (Supplementary Fig. 3d). These data are in line with the CUL3 expression profile in the human brain (Supplementary Fig. 3e–g) and point toward an important role for Cul3 in neuronal cells during early brain development. Thus, to understand whether behavioral defects are accompanied by neuroanatomical changes, we performed crystal violet (Nissl) stainings of adult brain sections obtained from Cul3 mutant and wild-type mice (Supplementary Fig. 4). Gross brain morphology appeared normal but we observed a slight reduction in cortical thickness (Supplementary Fig. 4b, c, e''). To investigate this neuroanatomical phenotype more closely, we stained the cortex, relevant for some of the behavioral phenotypes we observed, for Cux1 (upper layers 2/3) and Ctip2 (lower layers 5/6) (Fig. 3a–d). Quantifications in adult mice revealed that the Cul3 heterozygous mutation results in a mild decrease in upper and lower cortical layer thickness (Fig. 3b), a defect present already at birth (Fig. 3d). In addition, we found that the distribution of Cux1 and Ctip2-positive (Cux1+ and Ctip2+) cells is shifted toward lower cortical locations, indicative of abnormal cortical lamination (Fig. 3e).

In order to assess Cul3 dosage dependency as well as to underscore other potential subtle forebrain phenotypes, we decided to analyze also Cul3 null mice. Since constitutive homozygous deletion of Cul3 is embryonically lethal, we crossed conditional Cul3 animals ($Cul3^{fl}$) with an Emx1-Cre expressing line, generating forebrain-specific heterozygous and homozygous deletions of Cul3 ($Cul3^{+/fl}$ Emx1-Cre and $Cul3^{fl/fl}$ Emx1-Cre respectively) (Supplementary Fig. 5a–i). Importantly, Emx1-driven Cre expression[28] starts at E10.5, thus inducing Cul3 deletion at the beginning of forebrain development. While $Cul3^{+/fl}$ Emx1-Cre mice are viable and fertile, like $Cul3^{+/-}$ animals, $Cul3^{fl/fl}$ Emx1-Cre pups are much smaller than controls (Supplementary Fig. 5b) and die before weaning. In addition, $Cul3^{fl/fl}$ Emx1-Cre mice show severe brain malformations with pronounced cortical and hippocampal atrophy (Fig. 3f and Supplementary Fig. 5c). Similar to what we observed in $Cul3^{+/-}$ animals, fluorescence imaging of Cux1 and Ctip2 distribution revealed lamination defects in $Cul3^{fl/fl}$ Emx1-Cre and $Cul3^{+/fl}$ Emx1-Cre mice (Fig. 3g–i). Thus, proper Cul3 dosage is essential to guarantee correct brain development in mouse.

**Loss of Cul3 leads to neuronal migration defects in mice.** To identify the origin of the anatomical abnormalities observed in $Cul3^{+/-}$, $Cul3^{+/fl}$ Emx1-Cre, and $Cul3^{fl/fl}$ Emx1-Cre brains, we studied cell proliferation, apoptosis, and migration in the three different genotypes, focusing on the time window with the highest Cul3 expression (i.e., E14.5–16.5). Thus, we injected pregnant females with BrdU at E14.5 and collected the brains of the $Cul3^{+/fl}$ Emx1-Cre, $Cul3^{fl/fl}$ Emx1-Cre, and control embryos 2 h after injection. Brain samples were sliced and stained for BrdU, as indicator of cells in S-phase, and PHH3, a M-phase marker. Quantifications revealed a significant increase in PHH3+ and PHH3+BrdU+ double-labeled cells in $Cul3^{fl/fl}$ Emx1-Cre tissues at E14.5 (Supplementary Fig. 5d, e), suggesting a lengthening of the cell cycle in homozygous Cul3 mutant samples. In contrast, no differences were observed in $Cul3^{+/-}$ or $Cul3^{+/fl}$ Emx1-Cre mice (Supplementary Fig. 5d, e and Supplementary Fig. 6a). Considering the severe cortical atrophy observed at P0 (Fig. 3f), we extended our analysis to E16.5 BrdU-injected embryos. E16.5 samples were stained for Sox2, a neural stem cell marker; cleaved caspase-3 (cl. Casp3), a marker for apoptotic cells; and BrdU (Supplementary Fig. 5f–i). While in $Cul3^{fl/fl}$ Emx1-Cre we found a substantial increase of apoptotic cells in the developing cortex (of which 69.75% ± 1.47 were localized in the ventricular zone) and a corresponding reduction in Sox2+ and BrdU+ cells, we did not observe these anomalies in $Cul3^{+/fl}$ Emx1-Cre animals. Altogether, our data is in line with data from $Cul3^{-/-}$ cells, showing that homozygous deletion of Cul3 leads to mitotic arrest and cell apoptosis[18,29] and would explain the more severe cortical atrophy observed in $Cul3^{fl/fl}$ Emx1-Cre mice compared to heterozygous mutant animals.

The issues in cell cycle regulation upon homozygous loss of Cul3, however, cannot explain the thinning of cortical layers in the heterozygous mutant animals (i.e., $Cul3^{+/-}$ and $Cul3^{+/fl}$ Emx1-Cre). Therefore, we investigated a potential intermediate progenitor phenotype by staining for Tbr2 in $Cul3^{+/-}$ embryos at E14.5. Yet, we could not observe any difference in the number of

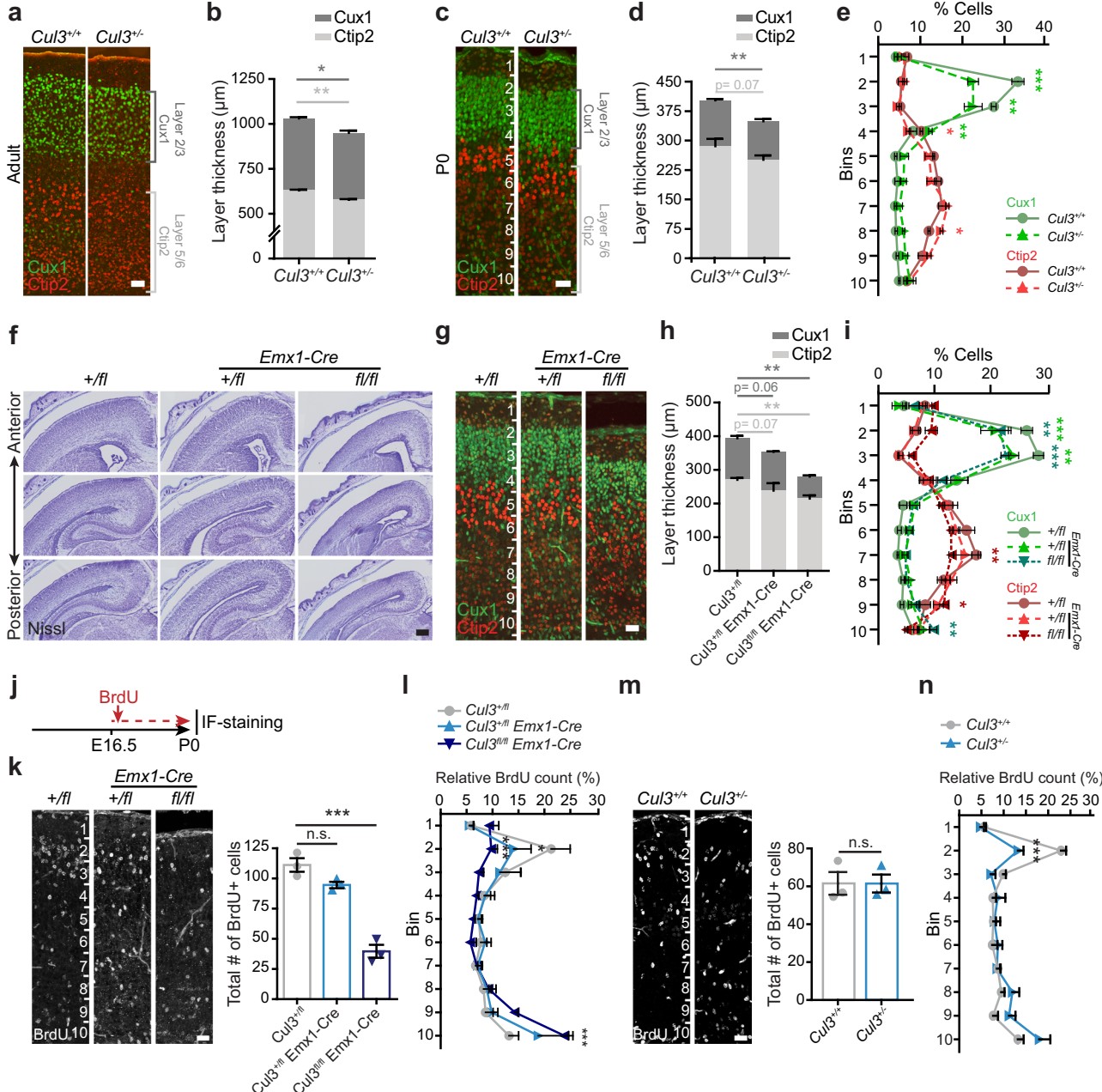

**Fig. 3 Abnormal lamination of the cortex and migration defects in *Cul3* mutant mice. a–d** Immunofluorescent stainings for Ctip2 and Cux1 on coronal brain sections revealed laminar thinning in adults (**a**, **b**) and newborn (P0) *Cul3*[+/−] animals (**c**, **d**) (*n*(adults) = 3 littermates per genotype; *n*(P0) = 6 littermates per genotype; *P* = 0.04; **P* < 0.01; 2-way ANOVA and Sidak's multiple comparison test). **e, i** Bin-wise comparison of relative cell numbers (in %) revealed a shifted Cux1/Ctip2 layer profile, indicating laminar defects at P0 (*n* = 3 littermates per genotype; *P* < 0.02, **P* < 0.01, ***P* < 0.001; 2-way ANOVA, Sidak's multiple comparison test). **f** Nissl-staining of P0 coronal, *Cul3*[+/fl], *Cul3*[+/fl] *Emx1-Cre*, and *Cul3*[fl/fl] *Emx1-Cre* brain sections show severe brain malformations in *Cul3*[fl/fl] *Emx1-Cre* pups (*n* = 3 littermates per genotype). **g, h** Immunofluorescent staining with antibodies against Ctip2 and Cux1 reveal cortical laminar thinning in both *Cul3*[+/fl] *Emx1-Cre* and *Cul3*[fl/fl] *Emx1-Cre* pups at P0 (*n* = 3 littermates per genotype; *P* < 0.05, **P* < 0.01, ***P* < 0.001; 2-way ANOVA, Sidak's multiple comparison test). **j** Scheme of the BrdU birthdate labeling experiments. **k, m** Injection of BrdU at E16.5 and anti-BrdU immunofluorescent (IF−) staining and analysis of total BrdU positive (BrdU+) cells in cortical columns at P0 shows severely decreased number of BrdU+ cells in *Cul3*[fl/fl] *Emx1-Cre* brains, but not in the *Cul3*[+/fl] *Emx1-Cre* and *Cul3*[+/−] cortex (*n* = 3 littermate pairs pups per genotype; ***P* = 0.0001, n.s. not significant; 1-way ANOVA and Sidak's multiple comparison test and unpaired two-tailed *t*-test). **l, n** Bin-wise analysis of relative numbers of BrdU+ cells showed decreased numbers of BrdU+ cells in upper bins and increased numbers of BrdU+ cells in lower bins in *Cul3*[+/fl] *Emx1-Cre*, *Cul3*[fl/fl] *Emx1-Cre* (**c**) and *Cul3*[+/−] (**e**) mice (*n* = 3 littermate pairs per genotype; *P* < 0.05, ***P* < 0.001; 2-way ANOVA and Sidak's multiple comparison test). Data presented as stacked bar-plots of mean ± SEM in **b**, **d**, **h**, and connected mean ± SEM in **e** and **i**. Data presented as mean ± SEM and scatter plot in k, m and connected mean ± SEM in **l** and **n**. Scale bars: 50 µm in **a**, 25 µm in **c**, **g** and 200 µm in **f**, 25 µm in **k**, **m**; numbers in **c**, **g**, **k**, **m** indicate depth in the cortex. Detailed statistics are provided in Supplementary Data 1.

intermediate progenitor cells in the SVZ of the heterozygous knockout (Supplementary Fig. 6b).

To test whether abnormal cell migration causes the lamination defects in heterozygous *Cul3* mutant mice we again pulsed E16.5 embryos with BrdU, but this time analyzed the number and position of BrdU+ cells in the cerebral cortex at P0 (Fig. 3j). We found a severe reduction in BrdU+ cells in *Cul3*^fl/fl^ *Emx1-Cre* P0 animals compared to control samples from *Cul3*^+/fl^ littermates (Fig. 3k), consistent with the above-mentioned increase in neural cell apoptosis in *Cul3*^fl/fl^ *Emx1-Cre* embryos. In addition, we found that a substantially smaller fraction of BrdU+ cells reaches the upper cortical layers and that a significant number of BrdU+ cells remain stranded in lower cortical layers in *Cul3*^fl/fl^ *Emx1-Cre* animals (Fig. 3l). These results suggest that complete deletion of *Cul3* in the forebrain leads to neural cell cycle defects, apoptosis and neuronal migration defects. Importantly, while *Cul3*^+/fl^ *Emx1-Cre* samples do not show a reduction in total BrdU+ cells, suggesting normal production and survival of cortical neural cells, *Cul3*^+/fl^ *Emx1-Cre* pups present a clear reduction of BrdU+ cells reaching the upper part of the cortex (Fig. 3l). We observed the same defect in the cerebral cortex of constitutive *Cul3*^+/−^ mice (Fig. 3m, n), indicating that *Cul3* haploinsufficiency is associated with a neuronal migration phenotype, thus explaining the observed lamination defects. Accordingly, when assessing the cell density in cortical layers V/VI, we counted a significant increase in the number of cell nuclei in $100 \times 100$ μm windows in layers V/VI of *Cul3*^+/−^ animals at P0, indicating that a substantial number of neurons accumulates in those layers (Supplementary Fig. 6c).

To obtain more direct evidence of migration defects in *Cul3*^+/−^ embryonic cortical tissues, we performed *in utero* ventricular injections of *human Synapsin-GFP*-encoding lentivirus at E13.5, prepared acute slices from transduced brains at E17.5, and performed live-cell imaging over the course of 16 h (Fig. 4a). Individual GFP-labeled radially migrating cells were manually tracked and the average migration speed and velocity were calculated for neurons in the *Cul3*^+/−^ and *Cul3*^+/+^ developing cortex. We found that radially migrating neurons in the *Cul3*^+/−^ cortex travel slower and shorter distances as compared to migrating neurons of their control littermates (Fig. 4b–f).

Next, since we had predominantly quantified the position of glutamatergic neurons (Fig. 3e, i), we tested whether other cell types in the brain might be similarly affected. To this end, we counted the density of interneurons, astrocytes, and microglia in the adult cortex. Interestingly, while the number of interneurons in the cerebral cortex is significantly reduced (Supplementary Fig. 6d), the amount and position, of astrocytes and microglia are unchanged in *Cul3* mutant animals (Supplementary Fig. 6e, f). This difference may be explained by the fact that *Cul3* expression is highest in excitatory and inhibitory neurons, potentially making them more susceptible to Cul3-dependent defective protein homeostasis (Supplementary Fig. 3d, g).

To better understand how *Cul3* mutations affect cell migration, we switched to an in vitro model system. Analysis of neural progenitor cells (NPCs) generated from E13.5 *Cul3*^+/+^ and *Cul3*^+/−^ cortices (Supplementary Fig. 6g) confirmed abnormal cell motility in vitro (Fig. 4g–j, Supplementary Fig. 6h). Specifically, we traced the movement of NPCs moving away from neurospheres over the course of several hours and found that *Cul3* mutant cells travel shorter distances than control cells (Fig. 4g, h). In addition, the live-imaging analysis revealed that mutant NPCs do not migrate far from the sphere, move less, and have reduced migration speed (Fig. 4i, j and Supplementary Fig. 6h, Supplementary Movies 1, 2).

**Cul3 haploinsufficiency leads to abnormal neuronal network activity.** We reasoned that defects in neural cell migration and cortical lamination could have an important impact on neuronal network activity. Indeed, other ASD-risk genes associated with defects in neuronal migration have been shown to substantially modify neuronal network activity in vivo[30]. To this end, we evaluated the spontaneous network activity in two-months-old *Cul3*^+/−^ mice, recording spontaneous postsynaptic currents (sPSC) from pyramidal neurons in layer 2/3 of the cortex in whole-cell configuration. Both spontaneous excitatory and inhibitory postsynaptic currents (sEPSC and sIPSC, respectively) are reduced in mutant animals (Supplementary Fig. 7a–f). In particular, *Cul3*^+/−^ mice show a reduction in sEPSC amplitude (Supplementary Fig. 7b) and frequency, evidenced by an increased mean inter-event interval (IEI) (Supplementary Fig. 7c) compared to wild-type littermates. *Cul3*^+/−^ and wild-type mice showed similar sIPSC peak currents, even if we notice a slight shift of the cumulative distribution towards higher amplitudes (Supplementary Fig. 7e), and mean frequency (Supplementary Fig. 7f). Of note, sIPSC distribution is shifted towards lower frequency (Supplementary Fig. 7f). Next, we calculated the ratio of sEPSC-to-sIPSC mean amplitudes and found that *Cul3* haploinsufficiency results in a reduced E/I ratio (Supplementary Fig. 7g), a circuit dysfunction occurring in a number of other ASD models[31–33].

To test whether the observed differences in neuronal network activity are due to morphological defects, we performed Golgi stainings and analyzed the morphology of layer 2/3 pyramidal neurons in adult *Cul3*^+/−^ mice (Supplementary Fig. 7h–j). However, neither the dendritic length, nor the number of dendrites or spines (Supplementary Fig. 7i), nor dendritic branching (Supplementary Fig. 7j) or spine morphology (Supplementary Fig. 7l) is altered in *Cul3* haploinsufficient mice. In line with comparable spine morphology, the G-actin/F-actin ratio was also unchanged in the mutant (Supplementary Fig. 7k). Altogether these results point to a tissue level reduction in network activity and global synaptic transmission, likely linked to the cortical lamination defects in *Cul3*^+/−^ mice.

**Whole-proteome analysis reveals abnormal amounts of cytoskeletal proteins in Cul3 mutant mice.** To gain insight into the molecular mechanisms underlying the observed defects, and in view of Cul3's E3 ubiquitin ligase function, we assessed the impact of *Cul3* loss on the global proteome of the developing forebrain in *Cul3*^+/−^, *Cul3*^+/fl^ *Emx1-Cre* and *Cul3*^fl/fl^ *Emx1-Cre* mutants. Protein extracts from dissected E16.5 cortices from control and mutant animals were analyzed by quantitative proteomics (Fig. 5a and Supplementary Fig. 8a). Analysis of the total proteome of *Cul3*^+/−^, *Cul3*^+/fl^ *Emx1-Cre,* and *Cul3*^fl/fl^ *Emx1-Cre* mutants, as well as corresponding controls, resulted in the identification of 8100 protein groups. For differential protein expression analysis, first, the differences between the two controls, *Cul3*^+/+^ and *Cul3*^+/fl^, were assessed and as these differences were minor (i.e., four proteins Wdfy1, Abca1, Ep400, and Cwf19l2, Supplementary Fig. 8b) a unique *Cul3*^ctrl^ data set was used for further analysis. Protein groups were then filtered based on fold change and False Discovery Rate (FDR) thresholds. Employing FDR thresholds of 10% we identified 31 up- and 33 down-regulated proteins in the *Cul3*^+/−^ embryonic cortex (Fig. 5b and Supplementary Data 3), and 38 up- and 22 downregulated proteins in the *Cul3*^+/fl^ *Emx1-Cre* embryonic cortex (Fig. 5c and Supplementary Data 4). As expected from the more severe phenotype of *Cul3*^fl/fl^ *Emx1-Cre* mutant pups, a much larger number of deregulated proteins were identified in conditional homozygous knockout embryos (146 up- and 94 downregulated

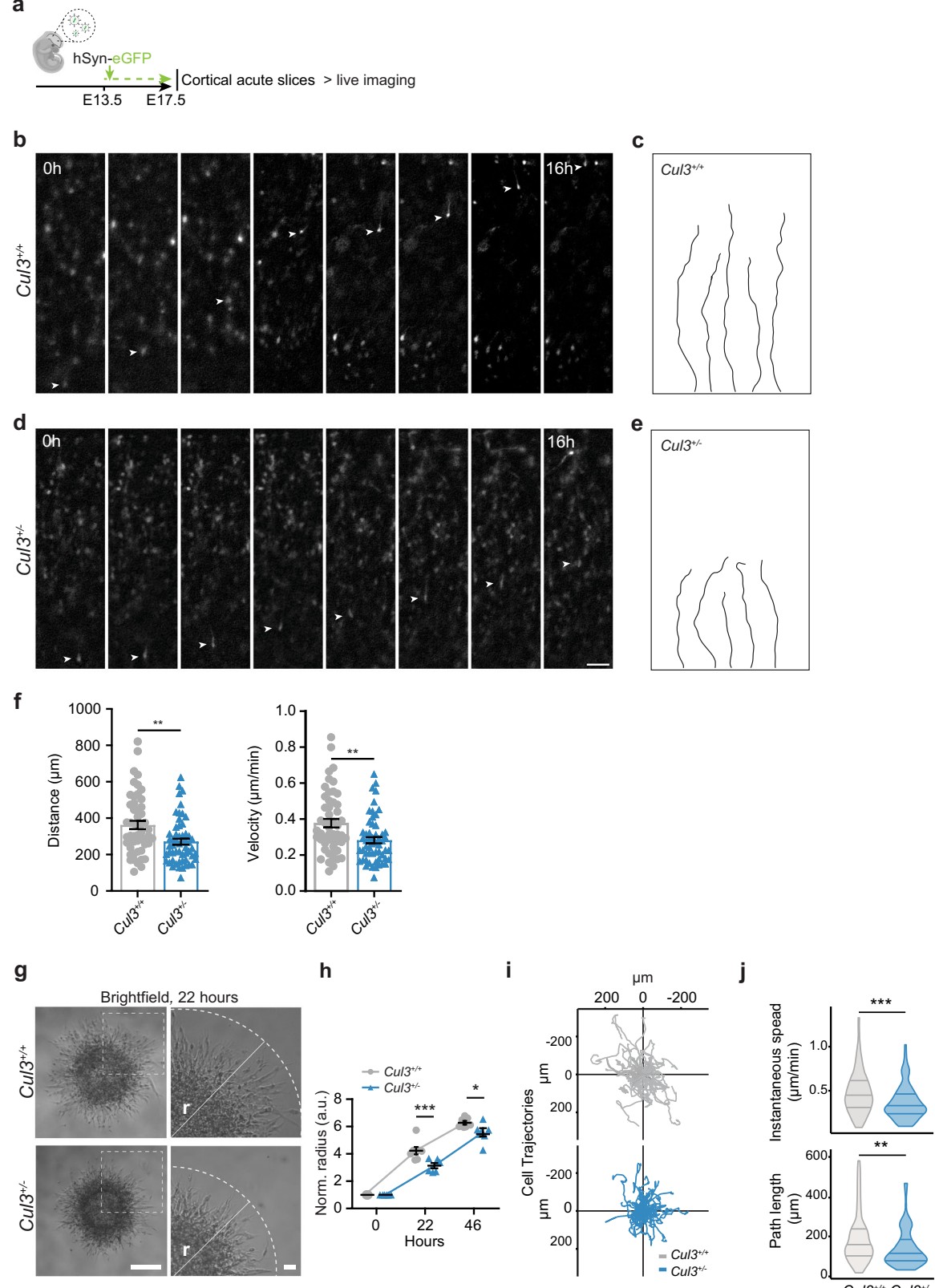

proteins, Fig. 5d and Supplementary Data 5). Overall, and in agreement with previous observations[34–36], the fold changes were mild, in line with the hypothesis that the ubiquitylated isoform of a protein represents only a small fraction of the total pool of that given protein at any time point[37]. The small number of deregulated proteins in the forebrain tissue of *Cul3* haploinsufficient embryos did not justify GO-term enrichment analysis. Therefore, to get an indication of the classes of proteins affected in *Cul3* mutants, we performed GO-term enrichment analysis on the *Cul3fl/fl Emx1-Cre* data at an FDR of 20%. We found that deregulated proteins were significantly enriched for DNA-directed RNA polymerase II core complex members and

**Fig. 4 Cul3 haploinsufficiency leads to migrations deficits in vivo and in vitro. a** Scheme of lentiviral labeling experiments followed by cortical acute slice preparation and time-lapse live-imaging. **b, d** Lentiviral injections of hSyn-eGFP at E13.5 and time-lapse imaging of migrating neurons over 16 h at E17.5 reveal migration deficits in the $Cul3^{+/-}$ cortex (arrowheads: eGFP-labeled cells); (n(animals) = 5 per genotype, $n(Cul3^{+/+})$ = 52 cells, $n(Cul3^{+/-})$ = 56 cells). **c,e** Representative cell trajectories of $Cul3^{+/+}$ and $Cul3^{+/-}$ cells indicate reduced path length upon $Cul3$ haploinsufficiency (**e**). **f** Quantification shows a significant decrease in cumulative distances and migration velocity in $Cul3^{+/-}$ cells (n(animals) = 5 per genotype, $n(Cul3^{+/+})$ = 52 cells, $n(Cul3^{+/-})$ = 56 cells; **P = 0.001; two-tailed Mann–Whitney U-test). **g, h** In vitro migration assay of matrigel embedded neurospheres generated from $Cul3^{+/+}$ and $Cul3^{+/-}$ NPCs reveals decreased migratory abilities (r = radius of the furthest migrated cell) 22 h (o representative images) and 46 h after plating. Radius was normalized to initial sphere size; (n(spheres) = 7/6 $Cul3^{+/+}$ and $Cul3^{+/-}$ respectively; *P = 0.011, ***P = 0.0008; 2-way ANOVA and Sidak's multiple comparison test). **i, j** Cell tracks of $Cul3^{+/+}$ and $Cul3^{+/-}$ NPCs detaching from the neurosphere into embedding bovine collagen matrix (n(spheres) = 3 per genotype, n(cells) = 30 per replicate) imaged in a single plane. Cell trajectories of each cell fixed at origin plotted in Euclidean plane (**i**). Mean instantaneous speed (**j** top) and total cell path length (**j** bottom) quantification (**P = 0.002; ***P = 0.0004; Wilcoxon rank-sum test). Data are shown as mean ± SEM and scatter plots in **f**, **h**. Data presented as mean ± SEM and violin plots with median and first and third quartiles (**j**). Scale bars: 50 μm in **d**, 200 μm in **g**, overview and 40 μm in **g**, close-up. Detailed statistics are provided in Supplementary Data 1.

proteins of the proteasome core complex. In addition, deregulated proteins were functionally linked to CNS and forebrain development, the regulation of cell migration, actin- and microtubule-cytoskeleton organization, cell–cell adhesion, and apoptosis (Fig. 5e and Supplementary Data 6). Given the migration phenotype observed in both heterozygous and conditional homozygous $Cul3$ mutant animals, we decided to investigate changes in cytoskeletal proteins further and found several of these to be misregulated also in $Cul3^{+/-}$ embryonic tissue (Fig. 5b–d in purple). We annotated these proteins manually, drawing on published literature and focusing on those appearing to follow a dose-dependent response to $Cul3$ loss, when the mean raw expression levels were fitted to a linear model ($Cul3^{+/fl} < Cul3^{+/fl}$ $Emx1$-Cre and $<Cul3^{fl/fl}$ $Emx1$-Cre) (Fig. 5f–h and Supplementary Data 7). Several have a putative or a confirmed function in cytoskeletal organization, and/or regulation of cell migration and differentiation. Some downregulated proteins belong to a family of microtubule-associated proteins linked to abnormal brain development in humans (i.e., Tubb2b, Map2)[38,39]. However, as Cul3 targets proteins for degradation, we were most interested in the identified up-regulated proteins, as their upregulation likely represents a direct consequence of $Cul3$ loss. To validate the results obtained by proteomic analysis we quantified by western blot two of the top up-regulated cytoskeletal-associated proteins observed in both $Cul3^{+/-}$ and $Cul3^{fl/fl}$ $Emx1$-Cre samples, Plastin 3 (Pls3) and Internexin neuronal intermediate filament protein A (INA) (Supplementary Data 7) as well as the cytoskeleton interacting protein Nischarin (Nisch) altered in $Cul3^{fl/fl}$ $Emx1$-Cre samples. Indeed, we confirmed that $Cul3$ deficiency leads to higher amounts of Pls3 and INA in both $Cul3^{+/-}$ and $Cul3^{fl/lf}$ $Emx1$-Cre E16.5 cortical lysates, as well as elevated levels of Nisch in $Cul3^{fl/lf}$ $Emx1$-Cre samples (Fig. 5i, j). Furthermore, we found that the upregulation of these proteins is not due to increased gene expression since there were no corresponding increases at the mRNA level (Fig. 5k), but rather to a translational or post-translational effect. Interestingly, $Pls3$ was actually downregulated at the mRNA level, likely a compensatory response to the accumulation of the protein, indicating a feedback loop to adjust the protein level of Pls3 and thus pointing towards a potentially important functional role of Pls3 in the brain.

In contrast to early development but in line with an important regulatory function of $Cul3$ mostly during early brain development, we found that in adult mutant cortical, hippocampal, and cerebellar tissues, $Cul3$ deficiency results in very few deregulated proteins (at 10% FDR, none in the cortex, hippocampus: 3 up- and 1 down-, and cerebellum: 2 up- and 1 downregulated proteins; Supplementary Fig. 8c–e and Supplementary Data 8). However, we found that Pls3 is also elevated in the juvenile and adult brain in constitutive $Cul3$ mutants (Supplementary Fig. 8f, g) as well as in tamoxifen-induced $Cul3$ haploinsufficient animals

(Supplementary Fig. 8h), further supporting a tight link between Pls3 and Cul3.

Recently, two studies exploring the role of $Cul3$ in the adult brain suggested a link between Cul3 and two distinct proteins, Smyd3[36] and eIF4G1[35]. Although the aims and methodologies employed are different from what used here, we analyzed the protein levels of Smyd3 and eIF4G1 in constitutive $Cul3$ haploinsufficient mice, which best resemble the human condition. Importantly, we could not observe changes in either Smyd3 or eIF4G1 in any of our proteomic data sets (Supplementary Fig. 8i,j). Accordingly, we did not find any change in eIF4G1 protein abundance in either the cerebral cortex, the hippocampus, or the cerebellum obtained from P14 male $Cul3^{+/-}$ pups (same time point, sex, and antibody as in Dong et al.[35]) (Supplementary Fig. 8k). In contrast, we found that Pls3 protein level is consistently increased in the $Cul3$ mutant mouse models employed by our colleagues[35,36], supporting our finding that Pls3 is an important substrate of the Cul3 ubiquitin ligase.

Taken together, our analysis points to a central role of Cul3 in the homeostatic regulation of cytoskeletal proteins of which the most significant appears to be Pls3.

**Pls3 regulates cell migration speed and actin organization.** Proteomic analysis of embryonic cortices highlighted an alteration of cytoskeletal protein levels in $Cul3$ mutant samples. In particular, Pls3 protein levels are consistently increased in all of our data sets. Therefore, we assessed whether Pls3 does play a direct role in cell migration. First, we employed the murine B16-F1 melanoma cell line, a routinely used cell line model to investigate cell migration of ectodermal cells[40,41]. To study the effect of $Pls3$ loss of function we employed a CRISPR-based system and generated three independent B16-F1 $Pls3$ knockout cell lines. As expected, CRISPR-Cas9-mediated homozygous disruption of the $Pls3$ locus completely abolishes $Pls3$ expression (Supplementary Fig. 9a). Excitingly, already in this cell line, in a random migration assay, $Pls3$ knockout cells show increased migration speed and traveled distance (Supplementary Fig. 9b,c), the exact opposite phenotype displayed by $Cul3$ mutant cells. Then, employing the above validated $Pls3$ sgRNA, we independently knocked out or overexpressed $Pls3$ in wild-type neural progenitor cells (Supplementary Fig. 9d) and tracked their movement over the course of 15 h. In support of a direct role of Pls3 in neural cell migration, we found that Pls3 knockout cells move faster than wild-type cells, while its overexpression leads to slower cell migration (Fig. 6a–c). Furthermore, we determined that the Pls3 effect on neural migration is cell-autonomous since transfected and un-transfected cells could be tracked side by side due to the presence of a green fluorescent protein (GFP) (Supplementary Fig. 9e). Taken

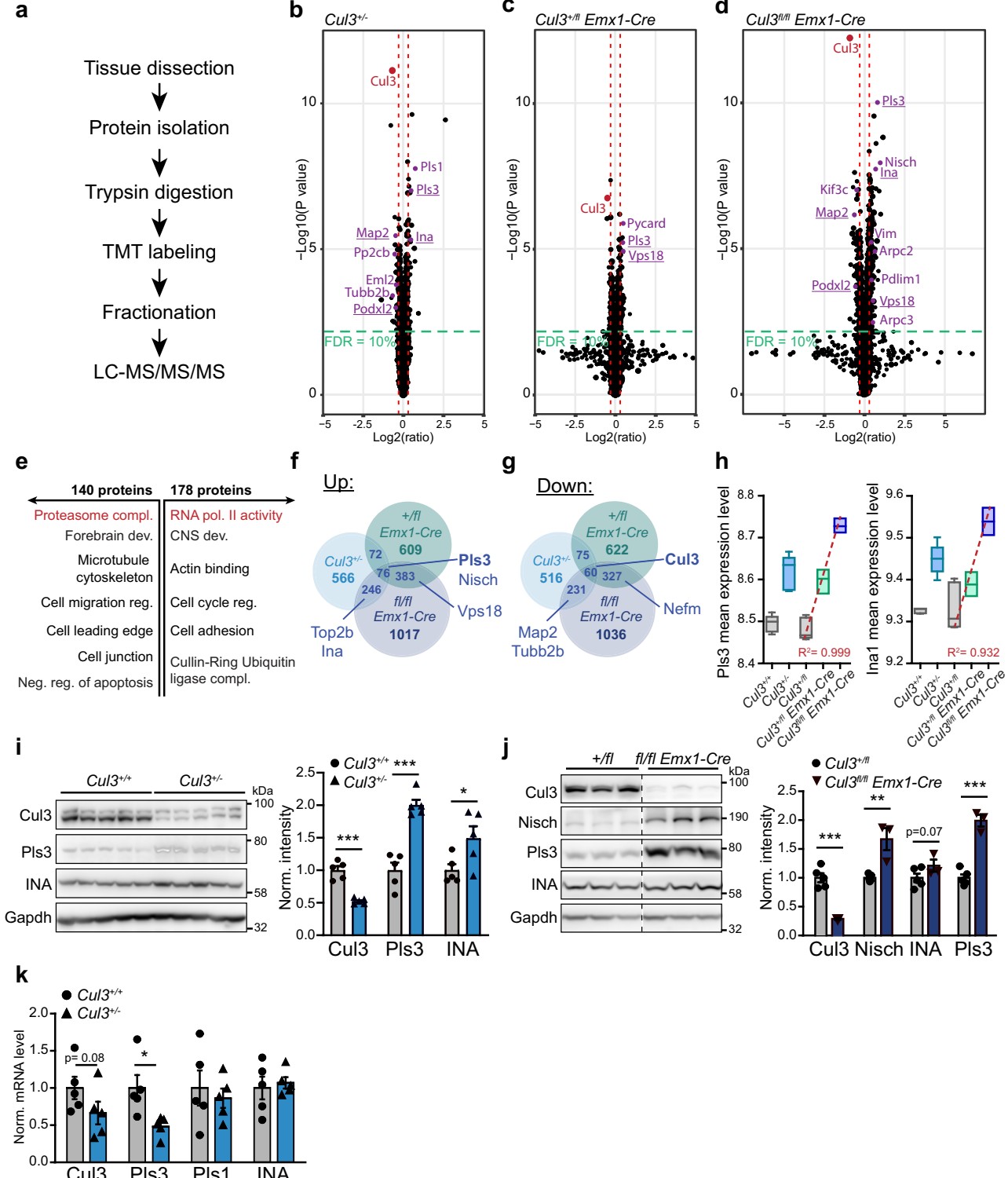

together our results indicate that Pls3 is an important player in the fine regulation of cell migration, and its level is inversely proportional to migration speed.

Although we clearly established that the amount of Pls3 protein is a determinant factor for cell migration dynamic, we went on and assessed whether the migration phenotype observed in *Cul3* mutant NPCs can be completely attributed to the elevated levels of Pls3 (Supplementary Fig. 9f). Thus, we knocked out *Pls3* in the context of *Cul3* haploinsufficiency (Supplementary Fig. 9d), and re-assessed neural cell migration. As expected, we found that

reducing *Pls3* levels is sufficient to correct the migration pattern observed in *Cul3*+/− cells (Fig. 6d–f).

Given that Pls3 is an actin-binding protein, we analyzed actin conformation in neural stem cells in the context of *Cul3* mutations[42], employing stimulated emission depletion (STED) super-resolution imaging. Specifically, we stained control and *Cul3* mutant NPCs with SiR-actin, a fluorescent probe for F-actin, as well as a tubulin antibody and analyzed actin filament orientation at the leading edge in diffraction-unlimited images (Fig. 7a). We found that *Cul3* deficiency leads to a disorganized

**Fig. 5 Deregulation of cytoskeletal proteins in *Cul3* mutant embryonic forebrain tissue. a** Sample preparation for proteomic analysis ($n(Cul3^{+/-}) = 5$ per genotype, $n(Cul3^{+/fl}\ Emx1\text{-}Cre) = 3$ per genotype, $n(Cul3^{fl/fl}\ Emx1\text{-}Cre) = 3$ per genotype; male mutant and control littermate pairs). **b–d**, Volcano plot of deregulated proteins at 10% FDR cut-off in the $Cul3^{+/-}$ developing cortex (**b**), $Cul3^{+/fl}\ Emx1\text{-}Cre$ (**c**), and $Cul3^{fl/fl}\ Emx1\text{-}Cre$ (**d**) developing forebrain; cytoskeleton related proteins (purple), proteins differently regulated in more than one data set are underlined, Cul3 (red) (**b–d**) (details: Supplementary Data 3-5). **e** DAVID functional annotation identified down- (left) and up- (right) regulated proteins at 20% FDR of the $Cul3^{fl/fl}\ Emx1\text{-}Cre$ forebrain involved in regulating the activity of RNA polymerase II, the proteasome core complex, neurogenesis and actin and microtubule cytoskeletal organization (selected GO-terms, significant GO-terms in red: RNA polymerase II core complex, GO: 0005665, adj. $P$-value = 0.002; proteasome core complex GO: 0005839, adj. $P$-value = 4.9e−07; details: Supplementary Data 6). **f, g** Comparison of ratios of up- and downregulated proteins in the $Cul3^{+/-}$, $Cul3^{+/fl}\ Emx1\text{-}Cre$ and $Cul3^{fl/fl}\ Emx1\text{-}Cre$ cortex at log 0.1 change (details: Supplementary Data 7). **h**, Fitting the mean raw expression levels of Pls3 and INA to a linear regression model indicates that these proteins follow a genotype-dependent dose-response (Pls3: $R^2 = 0.999$; INA: $R^2 = 0.932$; $Cul3^{+/fl} < Cul3^{+/fl}\ Emx1\text{-}Cre < Cul3^{fl/fl}\ Emx1\text{-}Cre$; boxplot shows the median value and 25–75th percentile, whiskers show minimum and maximum). **i** Western blot (left) and analysis of Gapdh normalized intensities (right) confirm increased levels of the cytoskeletal proteins Pls3 and INA in $Cul3^{+/-}$ lysates ($n = 5$ per genotype; *$P = 0.04$, ***$P < 0.001$; unpaired two-tailed $t$-tests). **j** Western blot and analysis of Gapdh normalized intensities confirm increased levels of cytoskeletal proteins Nisch, Pls3, and INA in $Cul3^{fl/fl}\ Emx1\text{-}Cre$ cortical lysates (samples on the same membrane cut for better visualization, $n(Cul3^{+/fl}) = 5$, $n(Cul3^{fl/fl}\ Emx1\text{-}Cre) = 3$; **$P = 0.003$, ***$P < 0.001$; unpaired two-tailed $t$-tests). **k** Quantitative real-time PCR analysis of mRNA levels of *Pls1*, *Pls3*, and *INA* normalized to wild-type levels ($\Delta Cq$ expression values; $n = 5$ per genotype; *$P = 0.02$; unpaired two-tailed $t$-tests). Data presented as or as box and whiskers, min to max (**h**) and mean ± SEM in **i–k**. Detailed statistics are provided in Supplementary Data 1.

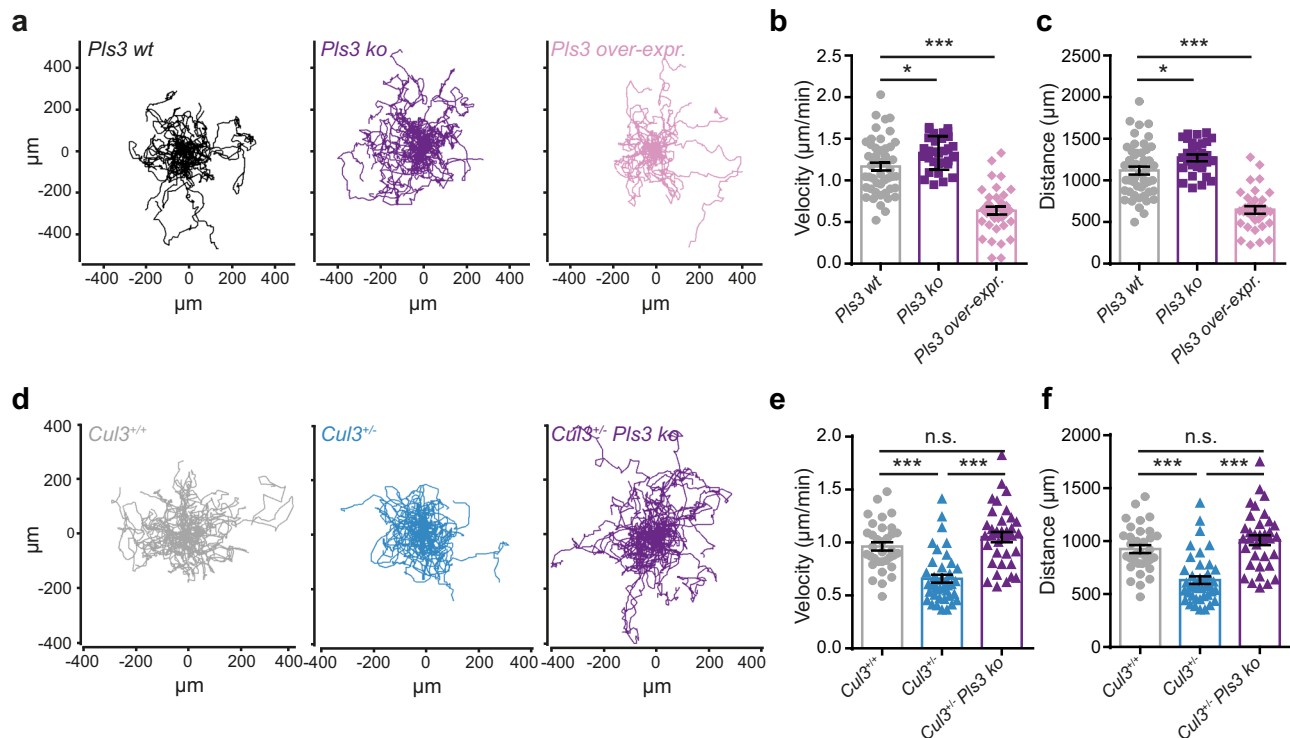

**Fig. 6 Pls3 regulates cell migration dynamics in *Cul3* haploinsufficient NPCs. a** 2D random migration assay was performed over a period of 15 h in *Pls3 wt*, *Pls3 ko*, and *Pls3 over-expressing* NPCs. Representative cell paths/trajectories of *Pls3 wild-type*, *Pls3 ko*, and *Pls3 over-expressing* cells indicate a larger migration area being covered by cells upon *Pls3 ko* (**a** middle), while over-expression of *Pls3* leads to a smaller migration area (**a** right). **b, c** Quantification revealed a significant increase in migration velocity and cumulative distance in *Pls3 ko* cells, and a significant decrease in *Pls3 over-expressing* NPCs relative to *Pls3 wild-type* cells ($n(Pls3\ wt) = 49$ cells, $n(Pls3\ ko) = 31$ cells and $n(Pls3\ over\text{-}expr.) = 37$ cells; *$P = 0.02$, ***$P < 0.0001$; 1-way ANOVA and Sidak's multiple comparison test). **d** 2D random migration assay in $Cul3^{+/+}$, $Cul3^{+/-}$ and $Cul3^{+/-}\ Pls3\ ko$ NPCs. Representative cell paths/trajectories show a smaller migration area covered by $Cul3^{+/-}$ NPCs (**d** middle) as compared to $Cu3^{+/+}$ cells (**d** left). Reducing *Pls3* levels in $Cul3^{+/-}$ mNPCs by *Pls3 ko* is sufficient to rescue the *Cul3* haploinsufficient phenotype (**d** right). **e, f** Quantification revealed a significant decrease in migration velocity and traveled distance in $Cul3^{+/-}$ NPCs, a phenotype rescued when downregulating *Pls3* levels through *Pls3 ko* ($n(Cul3^{+/+}) = 34$ cells, $n(Cul3^{+/-}) = 40$ cells and $n(Cul3^{+/-}\ Pls3\ ko) = 35$ cells; ***$P < 0.0001$; one-sided Kruskal–Wallis test and Dunn's multiple comparison test). Data are shown as mean ± SEM and scatter plots in (**b**, **c**, **e**, **f**). Detailed statistics are provided in Supplementary Data 1.

actin architecture at the cell front of adherent $Cul3^{+/-}$ NPCs (Fig. 7b–d), while the microtubule organization appears normal (Fig. 7e–g). Decreased directionality of actin cables and stress fibers may in principle be caused by an increased number of focal adhesion sites, as these puncta contain all possible angles and would thereby decrease the measured dominant direction. Thus, we further analyzed the number of adhesion points in

control and mutant cells. Surprisingly, adhesion site counting revealed a slight, yet significant, decrease in focal adhesions in the mutant cells (Supplementary Fig. 10a–c). A reduction of adhesion points is also in line with our proteomic data, which identified a reduction of cell adhesion proteins (such as Podxl, Nid2, and Emb). To understand whether increased levels of Pls3 alone are sufficient to change the actin architecture in

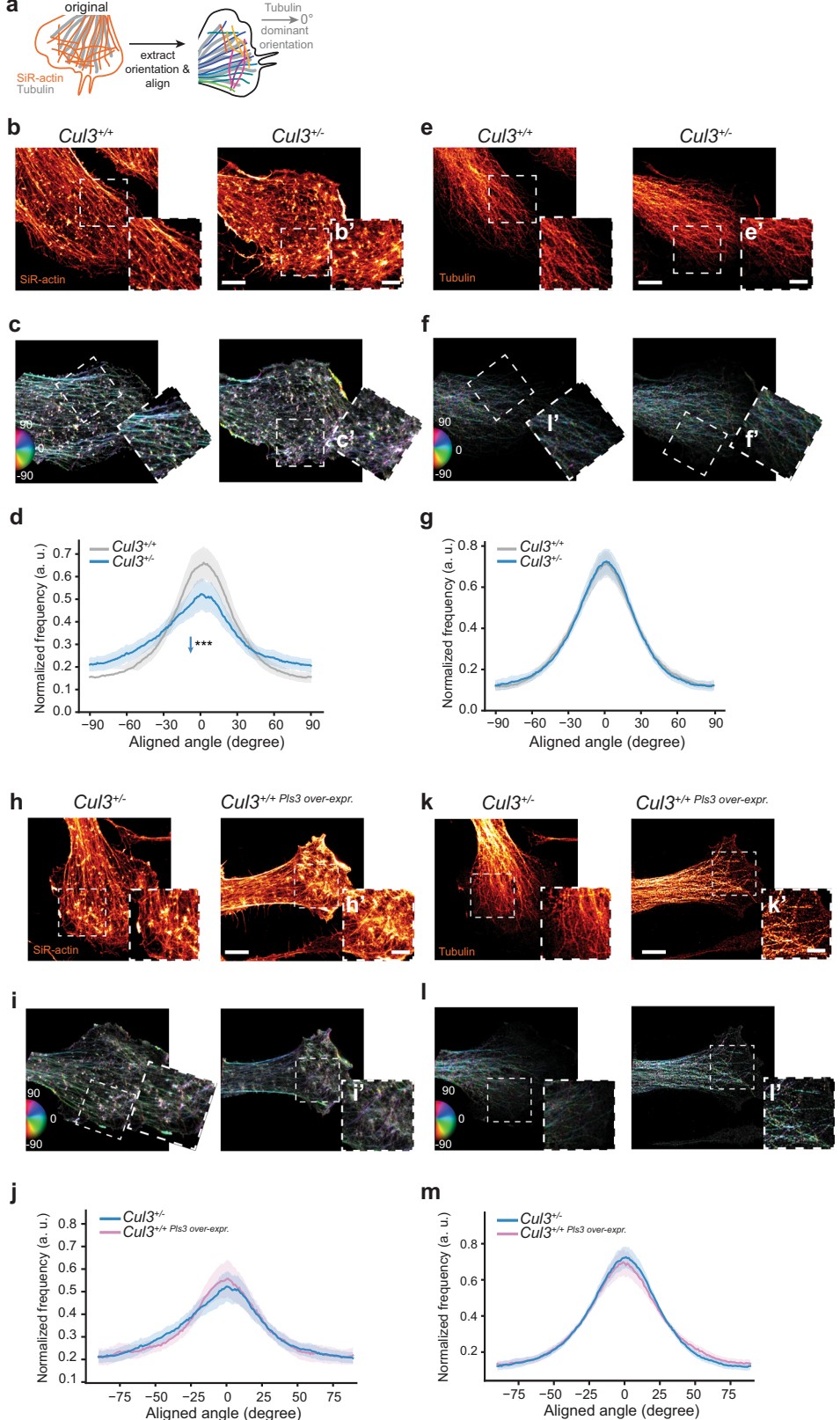

NPCs, we overexpressed *Pls3* in wild-type NPCs and analyzed their actin conformation by STED microscopy. Indeed, cells that overexpress *Pls3* in a wild-type background, show the same actin cytoskeletal organization as *Cul3*$^{+/-}$ NPCs (Fig. 7h–j and Supplementary Fig. 9g). Interestingly, *Pls3* over-expression additionally leads to an increase in focal adhesion counts

(Supplementary Fig. 10d–f), suggesting that in *Cul3*$^{+/-}$ additional compensatory, factors may come into play. Taken together, high-resolution imaging indicates a disorganized actin cytoskeleton at the cell front, probably constituting the underlying cell biological correlate of the migration defects observed in *Cul3* mutant cells.

**Fig. 7 Actin cytoskeleton disorganization in *Cul3* haploinsufficient and Pls3 over-expressing NPCs. a** Scheme of the actin orientation analysis at the cell front of NPCs. **b,e**, NPCs were stained using SiR-actin (**b**) and an anti-tubulin antibody (**e**). Cell protrusions were imaged employing STED-microscopy (close-up images in insets **b'**, **e'**). **c, f, i, l**, Processed, rotated and analyzed images, color code: hue as the orientation angle, saturation as coherency, and brightness represent photon counts in the STED image (close-ups in insets **c'**, **f'**, **i'**, **l'**). **d, g** Orientation distributions for actin (**d**) and microtubules (**g**) aligned to the dominant orientation angle of microtubules are shown for $Cul3^{+/+}$ and $Cul3^{+/-}$ cells ($n$(cells) = 43 per genotype from three independent NPC preparations; ***$P$ = 0.0003, n.s. not significant; two-tailed Welch's $t$-test). **h, k** $Cul3^{+/-}$ and *Pls3 over-expressing* NPCs stained using SiR-actin (**h**) and anti-tubulin antibody (**k**). Cell protrusions were imaged employing STED-microscopy (close-up images in insets **h'**, **k'**). **j, m** Orientation distributions for actin (**j**) and microtubules (**m**) aligned to the dominant orientation angle of microtubules are shown for $Cul3^{+/-}$ and $Cul3^{+/+}$ *Pls3 over-expressing* cells ($n$ ($Cul3^{+/-}$) = 43 cells), ($n$($Cul3^{+/+}$ *Pls3 over-expr.*) = 30 cells from three independent NPC preparations; n.s. not significant; two-tailed Welch's $t$-test). The dominant orientation is computed as the average orientation angle inside the cell. Plotted is the average angle distribution per group ± 95% confidence intervals. Scale bars: 5 μm in **b**, **e**, **h**, **k**, 2.5 μm in (insets **b'**, **e'**, **h'**, **k'**). Detailed statistics are provided in Supplementary Data 1.

***Cul3* transcription activation rescues cell migration defects.** While Pls3 may represent an optimal target to rescue *Cul3* haploinsufficiency-associated phenotypes there are no compounds currently available to reduce its activity. In addition, downregulating or blocking the function of an actin-binding protein may be harmful in vivo. Therefore, an alternative solution may be required.

Autism-associated gene variants are often de novo, heterozygous, loss of function mutations. This is indeed the case for all the *CUL3* ASD-associated mutations identified so far[1,2,4,6–11]. Thus, one potential therapeutic strategy could be to enhance gene expression from the non-mutated allele. In order to test this possibility, we made again use of the CRISPR system. This time, however, we expressed two independent sgRNAs targeting the promoter of the *Cul3* gene and a catalytically inactive Cas9 enzyme fused to a transcriptional activator (CRISPRa)[43] (Fig. 8a). First, we tested the ability of this system to increase *Cul3* expression in $Cul3^{+/-}$ NPCs by transfecting wild-type and mutant *Cul3* cells with the CRISPRa and either one of two guides targeting the *Cul3* promoter or a control guide (Fig. 8a) and performing real-time PCR to assess *Cul3* expression under the different conditions. Of note, both guides were able to enhance *Cul3* mRNA expression with one of the two guides (i.e., sgRNA 2) completely rescuing *Cul3* expression in $Cul3^{+/-}$ NPCs (Fig. 8b). In contrast, transfecting wild-type cells with either one of the *Cul3*-targeting sgRNA does not increase expression of the *Cul3* mRNA (Supplementary Fig. 11a), suggesting that other mechanisms may be in place to avoid Cul3 accumulation. Next, we tested neural cell migration under these conditions. As hypothesized, we found that increasing *Cul3* expression by leveraging on the intact allele is sufficient to rescue the neural cell migration defects presented by *Cul3* mutant cells (Fig. 8c–e). Importantly, we found that cells transfected with the *Cul3* specific RNA guides do not outrun wild-type cells but migrate exactly with the same speed and the same distance as in control conditions (Fig. 8c–e). In addition, enhancing *Cul3* mRNA levels is sufficient to normalize Pls3 protein levels in $Cul3^{+/-}$ cells, further underlining the tight regulatory link between Cul3 and Pls3 (Supplementary Fig. 11b). Altogether, our analysis shows that enhancing gene transcription of the wild-type allele is capable of rescuing phenotypes caused by ASD-gene haploinsufficiency and therefore may represent an optimal therapeutic strategy for this group of conditions.

## Discussion

De novo loF mutations in *CUL3* are an important cause of ASD, motor deficits, and intellectual disability in humans. Therefore, understanding the role of *CUL3* in the mammalian brain is of utmost importance. While the molecular function of CUL3 is well described, its role in the brain, particularly during development, remains largely unclear. To address these questions and model patients' *CUL3* haploinsufficiency, we analyzed a *Cul3* construct valid mouse model.

We found that constitutive *Cul3* heterozygous deletion leads to several behavioral abnormalities, including sociability issues, motor dysfunctions, and olfactory hyper-reactivity in adult animals. While this confirms the importance of maintaining correct *Cul3* dosage in the mammalian brain, it also implicates *Cul3* in the function of different brain regions, in line with the complex presentation of *CUL3* mutant patients. Importantly, for our behavioral studies, we invested effort in generating and analyzing male and female cohorts separately but we did not observe major phenotypic differences between genotypes of different sexes, supporting the observation that in humans *CUL3* mutations similarly affect males and females[1,2,8,10,11,43]. Thus, *CUL3* mutations do not appear to contribute to the skewed sex ratio in ASD cases.

While a gross morphological analysis did not reveal major abnormalities in the adult mutant brain, we found that *Cul3* haploinsufficiency leads to cortical lamination defects. Lamination abnormalities in $Cul3^{+/-}$ animals are linked to migration defects, which lead to retention of neuronal cells in lower cortical layers. *Cul3* mutation-associated neuronal migration defects are cell-autonomous and do not depend on brain-specific cues, as we also observe migration phenotypes in vitro. Furthermore, our data indicate that *Cul3* deficiency does not affect the number of astrocytes and microglial cells in the cortex, thus suggesting that *Cul3* regulates neuronal-specific processes, possibly due to its much higher expression in neuronal cell types compared to other brain cells types. Lamination defects, even when subtle, can have a profound effect on the physiology of the brain and disrupt the stereotyped organization of the microcolumnar structures typical of the neocortex. Accordingly, lamination defects have been previously associated with ASD in human and mouse models due to alteration of neuronal connectivity[44]. Similarly, *Cul3* haploinsufficient mice show decreased spontaneous activity of layer 2/3 cortical excitatory and inhibitory neurons, possibly due to a reduction of total neurons reaching these upper cortical layers and an overall incorrect laminar organization. Furthermore, when we calculated the E/I ratio, reflecting the interplay of the activity converging on an individual neuron, we found that *Cul3* haploinsufficiency results in a reduced E/I balance. Interestingly, this is in contrast with the work of Dong and colleagues[45], who observed an increased E/I ratio in pyramidal neurons of the CA1 in adult $Cul3^{+/fl}$ *GFAP-Cre* mice. This difference may be explained by the different brain regions analyzed, or by the mosaicism inherent to the used GFAP-Cre line, in which only 20–38% of the inhibitory neurons in the neocortex show Cre-expression. In any case, such circuit dysfunctions are a commonly described feature in a large number of other ASD models and are believed to underlie the ASD-relevant behavioral defects of these mouse lines, in agreement with the so-called "E/I imbalance hypothesis"[46,47].

Interestingly, complete deletion of *Cul3* leads to additional phenotypes including increased apoptosis during the neural stem

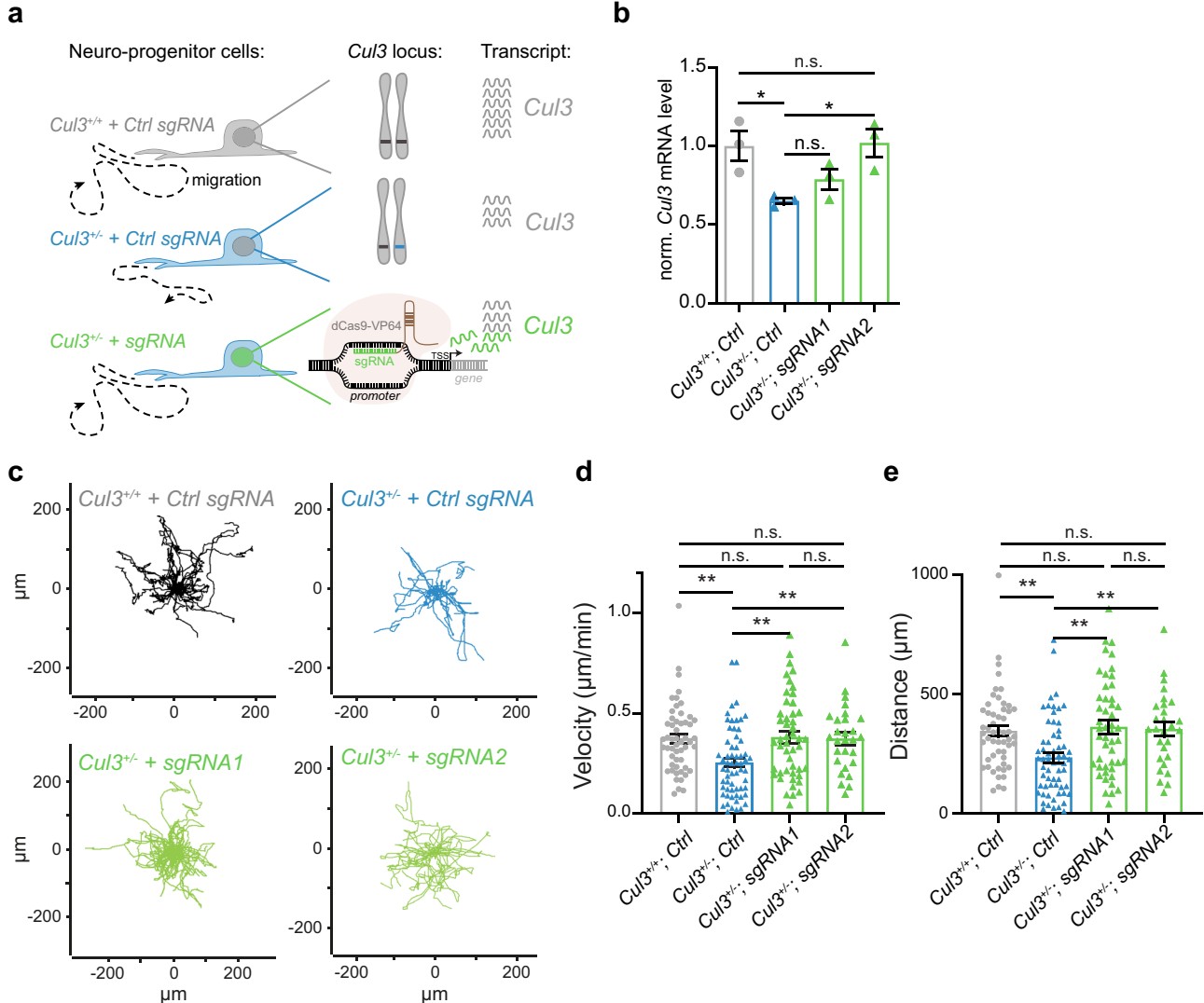

**Fig. 8 Transcriptional activation of *Cul3* rescues the cell migration defects displayed by *Cul3* haploinsufficient cells. a** Scheme of CRISPR-mediated activation (CRISPRa). While *Cul3*$^{+/+}$ + *Ctrl sgRNA* NPCs express wild-type mRNA levels and migrate physiologically (**a** top, gray), *Cul3*$^{+/-}$ + *Ctrl sgRNA* NPCs show reduced *Cul3* mRNA levels and are characterized by cell migration defects (**a** middle, blue). Upregulation of *Cul3* transcription, through a nuclease-deficient dCas9 fused to a transcriptional activator targeting the genomic *Cul3* promoter (**a** bottom, green), increases *Cul3* mRNA levels and thereby restores migration dynamics. **b** Quantitative real-time PCR analysis of *Cul3* mRNA levels normalized to the housekeepers *Gapdh* and *Pgk1* as well as the *Cul3* levels of wild-type NPCs transfected with their respective guides. Transfection of *Cul3*$^{+/-}$ cells employing two different sgRNAs sequences (*sgRNA 1* and *2*) targeting the *Cul3* promoter increased *Cul3* expression levels (ΔCq expression values; $n = 3$ per condition; *$P < 0.05$, n.s. not significant; 1-way ANOVA and Sidak's multiple comparison test). **c–e** 2D random migration assay was performed over a period of 15 h in *Cul3*$^{+/+}$ and *Cul3*$^{+/-}$ NPCs transfected with *Ctrl sgRNA*, as well as *Cul3*$^{+/-}$ cells transfected with *sgRNA1 and 2*. Representative cell trajectories indicate reduced migration by *Cul3*$^{+/-}$ + *Ctrl sgRNA* NPCs (**c** top right) as compared to *Cul3*$^{+/-}$ + *Ctrl sgRNA* controls (**c** top left). This *Cul3*$^{+/-}$ haploinsufficient phenotype is rescued through transcriptional activation of the *Cul3* promoter using two different sgRNAs (*Cul3*$^{+/-}$ + *sgRNA 1* and *2*; **c** bottom left and right). Quantification revealed a significant increase in migration velocity (**d**) and traveled distance (**e**) when restoring *Cul3* expression levels in transfected *Cul3*$^{+/-}$ NPCs ($n$(*Cul3*$^{+/+}$ + *Ctrl sgRNA*)= 53; $n$(*Cul3*$^{+/-}$ + *Ctrl sgRNA*)= 62 cells, $n$(*Cul3*$^{+/-}$ + *sgRNA1*)= 48 cells, and $n$(*Cul3*$^{+/-}$ + *sgRNA2*)= 28 cells; **$P < 0.01$, n.s. not significant; one-sided Kruskal–Wallis test and Dunn's multiple correction test). Data are shown as mean ± SEM and scatter plots in **b**, **d**, **e**. Detailed statistics are provided in Supplementary Data 1.

cell proliferation phase, possibly due to a defect in cell cycle progression. This is matched by a more severely affected proteostasis in the developing brain of Cul3$^{fl/fl}$ *Emx1-Cre* embryos. Defects of cell cycle progression were already associated with complete depletion of *Cul3* and do underlie the lethality of *Cul3* null mice[48]. Thus, studying construct valid *Cul3* models is critical to understand the bases of ASD in patients, as due to its stronger effect, homozygous deletion can obscure the underlying cellular and molecular drivers of *Cul3* haploinsufficiency-linked phenotypes.

A role of *Cul3* in cell migration was already hypothesized due to its connection with *Kctd13*, one of the genes involved in the 16p11.2 deletion syndrome, associated with neurological symptoms[49]. *Kctd13* encodes a substrate-linking protein for Cul3 and was suggested to bind RhoA, a regulator of the actin cytoskeleton, and thereby targeting it for ubiquitylation and degradation[49]. However, *Kctd13* deletion does not lead to elevated RhoA until after P7, and adult *Kctd13* heterozygous knockout mice do not have major structural brain differences as assessed by MRI[50]. Thus, in agreement with a lack of RhoA increase in

$Cul3^{+/-}$ embryonic forebrain tissue, as seen in our proteomics data, the observed migration and lamination defects are not driven by RhoA.

Importantly, while two recent studies[35,36] analyzed the effect of $Cul3$ deficiency and haploinsufficiency in the adult mouse brain, both studies fell short of studying the developmental consequences of $Cul3$ mutations in detail, employed Cre-lines that may miss critical developmental issues. In contrast, here we focused on the constitutive effects of $Cul3$ haploinsufficiency and discovered its crucial role in cell migration and proper organization of the cortex. Interestingly, however, when comparing our proteomic data set with these recently published studies we made a few remarkable observations. First, in mutant embryos, we found down-regulation of Map2, a microtubule-associated protein and known component of the neuronal cytoskeleton. Reduced levels of Map2 may be due to the reduced cortical layer thickness and numbers of neurons. Second, we found that the only protein consistently up-regulated in mutant animals is Pls3, under-investigated actin-bundling protein[51,52]. Pls3 is elevated in the developing and adult cortex, the adult hippocampus, but not in the adult striatum[35,36]. Thus, although there are some discrepancies, probably due to different experimental choices (e.g., mouse model employed, time point analyzed, etc.), the combined evidence suggests a role of $Cul3$ in regulating cytoskeletal organization by controlling Pls3 levels in the brain. In addition, our data shows a direct correlation between Cul3 activity and Pls3 levels, suggesting that Pls3 levels may be a good readout to test the pathogenicity of $CUL3$ mutations in humans. Interestingly, a pathogenic variant of $PLS3$ has recently also been described in a patient with idiopathic osteoporosis and ASD[53].

Our data indicate that Pls3 may act as a break of cell migration and that a tight regulation of its protein levels is critical to control cell-intrinsic migration properties. In the presence of low Pls3 levels, cells migrate faster and, consequently, travel longer distances, while in the presence of high Pls3 levels cells are moving slower and shorter distances. Abnormal homeostasis of Pls3 protein likely leads to the observed disorganization of actin architecture at the leading edge of migrating cells, thus explaining the migration defects displayed by $Cul3$ mutant cells in vivo and in vitro.

Altogether, our observations point to a central role of $Cul3$ in early brain development. In agreement, heterozygous deletion of $Cul3$ in adult animals does not cause obvious behavioral defects. Therefore, although a direct connection is still missing, it seems plausible that $Cul3$ haploinsufficiency-associated migration defects play a central role in the behavioral abnormalities associated with ASD. While some questions regarding the exact temporal trajectory remain to be answered, our findings point to a critical developmental time window for the emergence of $Cul3$ related, ASD-linked, behavioral abnormalities. Further, they suggest that later interventions may be ineffective in patients carrying mutations in the $CUL3$ gene. With regards to potential interventions, we show that reducing Pls3 levels in $Cul3$ mutant cells is sufficient to completely rescue the migration defects. Thus, drugs diminishing Pls3 activity may be therapeutically relevant for $CUL3$ patients. On the other side, we show that enhancing gene expression leveraging on the wild-type allele may represent a successful and more general strategy to correct phenotypes associated with ASD-gene haploinsufficiency.

In summary, our work provides novel insights into the pathophysiological and temporal basis of $Cul3$-linked behavioral abnormalities and offers important information on drug development and clinical trial design for this and other forms of ASD.

## Methods

**Mice**. We thank Dr. Jeffrey Singer (Portland State University, US) for providing us with the $Cul3^{flox}$ ($Cul3^{fl}$) conditional mouse line in which the exons 4 to 7 are flanked by loxP sites[18]. The $Cul3$ constitutive knockout mouse line ($Cul3^{+/-}$) was obtained by mating $Cul3^{flox}$ mice with a CMV-Cre line (B6.C-Tg(CMV-cre)1Cgn/J) and back-crossings to C57BL/6J wild-type animals (Supplementary Fig. 1a). Genotyping for the $Cul3$ knockout allele was performed using the following primers: *forward* GGAAACCTAAAGTTTTTATGCATG and *reverse* TTTGTCT GGACCAAATATGGCAGCCCAAACC. Details of all primers used can be found in Supplementary Table 1. The $Cul3$ Emx1-Cre conditional line was generated by crossing male $Cul3^{+/fl}$ mice with a Emx1-Cre expressing line (B6.129S2 (Emx1tm1cre)Krj/J). $Cul3^{fl/fl}$ Emx1-Cre mutant pups were sacrificed as soon as the phenotype was clearly detectable (strongly reduced size and very weak animals) within the first week of life, to comply with ethical requirements (3R principle). Embryonic time points were determined by plug checks, defining embryonic day (E) 0.5 as the morning after copulation. Animals were housed in groups of 3–4 animals per cage and kept on a 12 h light/dark cycle (lights on at 7:00 am), with food and water available *ad libitum*. All animal protocols were approved by the Institutional Animal Care and Use Committee at IST Austria and the Bundesministerium für Bildung, Wissenschaft und Forschung, Austria, approval number: BMWFW-66.018/0012-WF/V/3b/.

**Behavior**. All behavioral tests were performed on adult, 2- to 5-month-old, sex-matched littermate animals during the light period. Animal cohort sizes ranged between 8 and 20 animals per genotype and sex. Male and female cohorts were initially analyzed separately, and data was only pooled in case wild-type male and wild-type female data was comparable. Before testing, animals were habituated at least for 1 h to the testing room, equipment was cleaned with 70% EtOH after each animal. Different behavioral tests in the same mouse cohort were separated by at least one day of break. Tests were performed starting with the least aversive task and ending with the most aversive, and either scored automatically or by an experimenter blind to the genotype.

*Hind limb clasping*. During a 10 s tail suspension period hind limb clasping severity (scores 0–1- no hind limb clasping to 3- most severe phenotype) was assessed by the experimenter. The test was repeated three times per animal and the average score was calculated, scores between 0 and 1 are not considered as hind limb clasping.

*Gait analysis*. Gait properties were assessed by footprint analysis as described previously[20]. Briefly, fore- and hind paws were colored with non-toxic dye and the animals allowed crossing a white sheet of paper, 70 cm in length, in a straight line. Stride, sway, and stance length, as well as paw distance, were measured as indicated in Fig. 1c by the experimenter.

*RotaRod*. Performance on the accelerating mouse RotaRod (Ugo Basile S.R.L, Cat. No. 47650) was analyzed on 3 trials per day over three consecutive days, with an acceleration from 5 to 40 rpm over 5 min (300 s). For each animal the latency to fall (sec) and maximum speed at the end of each trial (rpm) were automatically determined, sex-matched littermates were tested simultaneously to avoid confounding factors. Initial coordination was assessed by analyzing the first trial on the first day of training.

*Open field*. As previously described[54], animals were allowed to freely explore a brightly lit arena (45 × 45 × 30 $cm^3$), made out of gray Plexiglas, over a 30 min time period. Locomotor activity (distance traveled and velocity) and center crossings were recorded by a video camera and analyzed using the EthoVision XT software (v.11.5; Noldus).

*Elevated plus maze (EPM)*. Mice were placed in the center of the EPM apparatus facing the open arm and left to explore the maze for 6 min. The time spent in the open and closed arms along with the number of closed and open arm entries and distance traveled in each arm were determined using Ethovision XT (v.11.5) video tracking system and software. The ratio of the time (in s) spent on the open arm vs. the total time spent on the maze was calculated.

*Three-chamber sociability test*. Mice were tested for sociability and social novelty preference as described previously[20]. The testing apparatus was a rectangular clear Plexiglas three chambers box (60 cm (L) × 40 cm (W) × 20 cm (H)). The dividing walls had doorways allowing access to each chamber. Age and sex-matched C57BL/6J mice were used as stranger mice and were habituated to placement inside the wire cage. Each test animal was first placed into the center chamber with open access to both left and right chamber, each containing an empty round wire cage. The wire cage (12 cm height, 11 cm diameter) allows nose contact between mice but prevents fighting. After 10 min of habituation, during the social phase, an age-matched stranger was placed in the left chamber while a novel object was placed into the right chamber. The test animal was allowed to freely explore the social apparatus for 10 min. Subsequently, each mouse was tested in a second 10 min session to evaluate the preference for a novel stranger, which was placed

inside the right wire cage. The number of nose contacts (<5 cm proximity) with the caged mouse, as well as the time spent in each chamber, was calculated. Analysis was done using the Noldus EthoVison XT (v.11.5). In addition, the social preference/novelty index for each mouse was calculated, as described previously[55].

*Olfaction habituation and dishabituation test (OHDH).* The test was performed as previously described[26]. In brief, mice were presented with a sequence of non-social, i.e., water, almond (McCormick) and banana (McCormick) (diluted 1:100 in water), and social cotton swabs (Social A and B). Social odors were obtained by wiping the cotton swabs in a zick-zack manner through the bedding of two distinct cages, housing each three, to the test mice, sex- and age-matched wild-type mice. Each cotton odor swab was presented for 3 × 2 min with a 1 min break. The time tested mouse spent sniffing each cotton swab was determined by the experimenter using a stop swatch. The adapted ODHD, Fig. 2h, was performed as described above, however excluding the non-social odors almond and banana.

*Contextual fear conditioning (CFC) task.* As described previously[54], mice were subjected to the CFC task in three sessions, each distanced by 24 h: a training session (day 1) and two re-exposure sessions (day 2 and day 3). On day one, mice were subjected to a single fear conditioning training session of 5 min, in which they learned to associate the conditioned stimulus (CS: context) to the unconditioned stimulus (US: a foot-shock). To this end, each mouse was placed in a fear-conditioning chamber (18 × 18 cm$^2$, Noldus) with an electrified grid floor. After 120 s of free exploration, the mouse was subjected three times to a foot-shock (0.5 mA; 2 s) delivered through the grid-floor, every 1 min. Mice remained in the conditioning chamber for 1 min after the last shock delivery. On days two and three, each mouse was placed back in the conditioned chamber for 10 and 5 min, respectively, without delivering the electrical foot-shock to test their memory retention and memory extinction. Behavior during all experimental sessions was recorded by a video camera mounted above the ceiling of the cage and connected to a computer equipped with the Ethovision XT software (Noldus). Percentage of time spent freezing (absence of all but respiratory movements for at least 3 s) was scored to assess emotional reactivity during training (day 1) and fear memory during retention test (the first 3 min of day 2 session) and extinction test (first 3 min of day 3 session). All behavioral parameters were scored by an experimenter blind to the animal experimental condition. Pairs of control-mutant littermates were randomly tested in the morning and in the afternoon to control for circadian rhythm.

## Immunofluorescence; BrdU labeling and imaging

*Immunofluorescent staining in adults.* Adult male littermate mice were deeply anesthetized and transcardially perfused with 4% paraformaldehyde (PFA). Brains were dissected, postfixed in 4% PFA, dehydrated in 30% sucrose, and sliced at 40 μm on a sliding VT1200S vibratome (Leica Microsystems). Stainings on adult brains were performed on floating sections without antigen retrieval. In brief, sections were washed in 1× PBS and incubated overnight on a horizontal shaker at 4 °C in primary antibody solution (14–16 h). Primary antibodies were diluted in 0.3% Triton X-100 and 2–5% donkey serum. On the next day, the sections were washed and incubated with a species-specific secondary antibody for 2 h at 4 °C. Nuclear counterstain was performed for 10 min with 300 nM DAPI (Life Technologies) in 1× PBS before mounting in DAKO fluorescent mounting medium. To examine cortical layering, the thickness of Cux1-positive cell layer and the Ctip2-positive cell layer was measured at three defined points of each cortical hemisphere (n = 3 littermate animals per genotype, at least 4 images/animal). For interneuron, microglia, and oligodendrocyte counting, positive-stained cells were counted within the somatosensory cortex and normalized to the area used (n = 3 littermate animals per genotype, at least 5 images/animal). Adult cortical sections were stained with the following primary antibodies: anti-Cux1 (Santa Cruz, sc-13024, 1:200), anti-Ctip2 (Abcam, ab18465, 1:500), anti-Parvalbumin (1:500, Chemicon MAB1572), anti-Iba1 (1:500, Wako 019 19741), anti-GFAP (1:200, Cell Signaling 12389 P).

*Immunofluorescent staining in embryos and newborn pups (P0/P1).* Mice were killed for analysis at E14.5 and E16.5 and P0 by decapitation. Heads were drop fixed in 4% PFA overnight, dehydrated in 30% sucrose, embedded in O.C.T. (Tissue Tek) and 18 μm sections were prepared on a Microm HM560 cryostat (Thermo Scientific). For assessment of the cellular composition of the embryonic cortex by immunofluorescent stainings, antigen retrieval was performed using 1× DAKO antigen retrieval solution (s1699) and immuno-fluorescent staining was performed as outlined above. The following primary antibodies were used: anti-Cux1 (Santa Cruz, sc-13024, 1:500), anti-Ctip2 (Abcam, ab18465, 1:500), anti-cleaved Caspase-3 (Cell Signaling, 9661, 1:300), anti-PHH3 (Millipore, cat. nr. 06-570, 1:500), anti-Sox2 (Millipore, AB5603, 1:200), anti-Tbr2 (Abcam ab183991, 1:250). Analysis of Cux1/Ctip2 cell distributions at P0 was performed by counting the relative number of Cux+ and Ctip2+ cells in each of 10 bins (bin height adjusted to cortical thickness, bin width 200 μm) in the cortex, normalized to the total number of positive cells in the respective region (n = at least 3 littermate mice per genotype, at least 4 images/animal).

*BrdU-based birthdate labeling for migration and proliferation analysis.* For cell cycle analysis, pregnant mice were injected with 0.1 mg/g bromodeoxyuridine (BrdU) at E14.5 and E16.5 and sacrificed 2 h later; embryos were decapitated and processed for immunostaining as described above. Cells in S-phase that incorporated BrdU were detected using anti-BrdU (BioRad, MCA2060T, 1:500). The number of BrdU+ cells was manually counted in cortical regions of 200 μm width in blinded images (n = 3 littermate animals per genotype, at least 5 images/animal). For migration analysis, pregnant females were injected with BrdU at E16.5, as described elsewhere[56]. Upon delivery (P0), pups were decapitated and tissue was processed for immunostaining as outlined above. For migration analysis, sections were stained with anti-BrdU (BioRad, MCA2060T, 1:500), and anti-Ctip2 (Abcam, ab18465, 1:500) antibodies. The P0 cortex was divided into ten bins of the same size and the relative number of BrdU+ nuclei per bin counted manually in blinded images (n = at least 3 littermate mice per genotype, at least 5 images/animal).

*Imaging.* Images from immunofluorescent stainings were acquired on a Zeiss LSM800 inverted confocal microscope using ZEN Blue imaging software (v.2.3), background corrected and adjusted for contrast and brightness, as well as analyzed in Fiji (v1.52n)[57] using the cell counter plugin.

## Nissl staining and Golgi staining

*Nissl staining.* For Nissl staining, brains from perfused adult animals were post-fixed in 4% PFA overnight, dehydrated and paraffin-embedded. Sagittal and coronal sections were cut on a Microtome HM 355 at 10 μm thickness. For Nissl stainings in newborn mice, pups were decapitated at P0, brains dissected, drop fixed in 4% PFA, dehydrated in sucrose, embedded in O.C.T, and cut at 18 μm on a cryostat. Nissl staining with 1% Cresyl Violet solution (Cresyl Violet Acetate, Sigma, Cat.No C 5042) was performed upon clearance of paraffin slices with RotiHistol (Carl Roth) for 10 min and rehydration of sections (absolute EtOH to water: 96%, 90%, 70%, 50%, 30%, water, 3–5 min each), or 3 × 5 min washes in 1× PBS to remove the OCT. Nissl stainings of adult and P0/P1 brains were captured using an Olympus Slide scanner VS120 and analyzed using Fiji (v1.52n).

*Golgi staining and analysis.* Golgi-Cox staining was performed according to protocol using the FD Rapid GolgiStain Kit$^{TM}$ (FD Neurotechnologies). After three weeks of Golgi impregnation, brains were cut coronally (120 μm) using a Leica Vibratome (Leica VT 1200 s) and mounted onto 1% gelatin-coated slides. Slides were then dehydrated through graded ethanol steps, cleared with RotiHistol (Carl Roth), and mounted with DPX mounting medium on coverslips (#1.5). To quantitatively analyze pyramidal neurons in Golgi-stained slides, impregnated pyramidal cells (8–10 neurons per brain, n = 3 littermate brains per genotype) of layer 2/3 in the somatosensory cortex were selected and imaged with a Nikon Eclipse Ti2 using a ×40 magnification. For analysis, single pyramidal neurons were manually reconstructed using Imaris analysis software (version 9.3.1). The average filament area, filament length, and Sholl intersections were analyzed using the same software. Spine counting was performed using Fiji (v1.52n), spines that started from 100 μm distance of the apical dendrite were counted within a 100 μm segment.

## Ex vivo live-imaging of acute cortical slices

*Lentiviral production.* Lenti-X 293T cells (Takara, cat. nr. 632180) used for virus production were grown in DMEM (Sigma cat. nr. D5796) at 37 °C and 5% CO$_2$. Briefly, eGFP-hSyn Lentivirus was produced by co-transfecting Lenti-X 293T cells using polyethylenimine (PEI) with 30 μg of hSyn-eGFP plasmid (plasmid was generously provided by Yoav Ben Simon, IST Austria), 18 μg of packaging plasmid pPax2 (Addgene plasmid ID: 12260) and 7.5 μg of envelope plasmid pMD2.G (Addgene plasmid ID: 12259) The next day, the culture medium was exchanged and cell supernatant was collected 24 h after transfection and filtered through 45 μm filters (Millipore). The virus-containing supernatant was overlaid on a 20% sucrose-containing buffer (20% sucrose, 100 mM NaCl, 20 mM HEPES (pH 7.4), 1 mM EDTA (pH 7.9) in sterile water) at a 5:1 v/v ratio, transferred to an ultra-centrifuge (Thermo Scientific, Sorvall WX) and spun at 112,000 × g for 1.5 h at 4 °C. Pellets were re-suspended in 200 μl PBS and aliquots were stored at −80 °C until used. To determine the viral titer, Lenti-C 293T cells were transduced with serially diluted (0.1; 0.01, 0.001) viral suspensions and the percentage of GFP+ cells was determined by FACS analysis using a BD FACS Canto II Analyzer. The viral titer was in the range of 2–3 × 10$^8$ TU/μl.

*Animal surgery.* Pregnant mice were anesthetized at E13.5 with isoflurane in combination with a subcutaneous injection of Metacam (0.005 mg/g) as an analgesic, mice were kept under constant isoflurane administration through a nose cone. After exposing both uterine horns, 2 μl of lentiviral suspension mixed with 0.1% Fast Green was injected through the uterine walls into the embryonic lateral ventricle as described in[58]. Following injections, the uterus was carefully relocated into the peritoneal cavity and the muscular walls and overlying skin were sutured. Following 4 days of lentiviral expression, brains were collected at E17.5 for slice culture preparation.

*Acute slice culture preparation and live-imaging of embryonic brains.* Brain slices were obtained from E17.5 *Cul3+/−* and wild-type littermates. Embryos were decapitated, whole brains were rapidly removed from the skull and quickly dissected in ice-cold cutting solution, containing (mM): 93 NMDG, 2.5 KCl, 1.2 NaH$_2$PO$_4$, 30 NaHCO$_3$, 20 HEPES, 25 glucose, 5 sodium ascorbate, 2 thiourea, 3 sodium pyruvate, 10 MgCl$_2$, 0.5 CaCl$_2$ (320 mOsm, 7.2–7.4 pH). Acute coronal slices (250 µm) were sectioned using a McIlwain Tissue Chopper (Mc Ilwain, Redding, USA) and recovered for 1 h at 4 °C in regular artificial cerebrospinal fluid (ACSF), containing (mM): 125 NaCl, 2.5 KCl, 1.25 NaH$_2$PO$_4$, 25 NaHCO$_3$, 25 glucose, 1 MgCl$_2$ and 2 CaCl$_2$ (320 mOsm, 7.2–7.4 pH); supplemented with Hoechst (Invitrogen, 1:5000). The cutting solution and ACSF were oxygenated for at least 30 min prior to dissections with 95% O$_2$ and 5% CO$_2$ to maintain the physiological pH.

Subsequently, slices were positioned onto 0.4 µm porous Millicell inserts (Millipore), placing 3–5 cortical slices onto each membrane. Membranes were then placed into 6-well plates with bottom #1.5H glass coverslips (MatTek) immersed with 1.5 ml of slice culture medium/well (50% HEPES-buffered MEM (Thermo Scientific), 25% HBSS (Thermo Scientific), 25% Heat Inactivated Horse-Serum (Thermo Scientific), and 1mM L-Glutamine (Thermo Scientific)). For recovery prior to imaging, slice cultures were placed at 37 °C with 5% CO$_2$ for 30 min.

GFP time-lapse images were acquired with a ×10 objective on an inverted Nikon Ti2 microscope at 37 °C with 5% CO$_2$ for 16 h every 15 min with Z stacks (Z-step 15 µm). Velocity and migration patterns of individual GFP-labeled neurons were tracked manually employing the built-in Fiji plugin Manual Tracking. Data of traced cells were then imported into the Chemotaxis and Migration Tool (Ibidi, https://ibidi.com/chemotaxis-analysis/171-chemotaxis-and-migration-tool.html), analyzing tracks of individual cells, quantifying the track velocity and cumulative distance.

## LC-MS/MS/MS whole-proteome analysis

*Samples.* Adult male *Cul3+/−* and *Cul3+/+* wild-type littermate animals (cortex and hippocampus: 4 littermates per genotype; cerebellum : 5 littermates per genotype) were deeply anesthetized and transcardially perfused with 15 ml of ice-cold 0.9% NaCl to clear the brain from blood. The cortex, hippocampus, and cerebellum were rapidly dissected on ice, snap-frozen in liquid nitrogen, and stored at −80° until protein extraction. Embryonic E16.5 forebrain tissue (pooled developing cortex and hippocampus; male embryos *n*(constitutive KO) = 5 mice per genotype, *n*(Emx1Cre conditional KO) = 5 *Cul3+/fl*, 3 *Cul3+/fl Emx1-Cre*, and 3 *Cul3fl/fl Emx1-Cre* respectively; for batch correction purposes each litter selected contained at least one control animal, see Statistical Data Analysis) were dissected on ice, meninges were removed and samples were snap-frozen in liquid nitrogen and stored at −80° until protein extraction. Tissues were homogenized 1:5 (w:v) in modified RIPA buffer (50 mM Tris-HCl pH 7.5, 150 mM NaCl, 1% NP40, 0.5% Sodium deoxycholate, 0.1% SDS, 1 mM EDTA, 10 mM NaF) supplemented freshly with protease and phosphatase inhibitors (Roche 04 693 159 001 and 04 906 837 001) and lysed for 30-45 min on ice, while occasionally being vortexed every 10 min. Each sample was sonicated twice at 180 W in an ice-cold water bath and centrifuged at 12,000×*g* for 20 min at 4 °C. Lysates were quantified using the Pierce™ BCA Protein Assay Kit (ThermoFisher, Cat. no. 23225).

*TMT labeling and High pH reversed-phase chromatography.* Aliquots of 100 µg of each sample were digested with trypsin (2.5 µg trypsin per 100 µg protein; 37 °C, overnight), labeled with Tandem Mass Tag (TMT) 11plex reagents according to the manufacturer's protocol (ThermoFisher Scientific, Loughborough, LE11 5RG, UK) and pooled. For the adult data set, where the number of samples exceeded the number of available TMT channel, one combined TMT sample was generated for each tissue. For the embryonic data set, one TMT sample contained *Cul3+/+* and *Cul3+/−* samples, while the other sample contained the *Cul3+/fl*, *Cul3+/fl Emx1-Cre*, and *Cul3fl/fl Emx1-Cre* samples. Details of TMT labeling can be found in Supplementary Data 9 (adult) and 10 (embryonic). Pooled samples were evaporated to dryness, re-dissolved in 5% formic acid, and then desalted using a SepPak cartridge according to the manufacturer's instructions (Waters, Milford, Massachusetts, USA). The eluate from the SepPak cartridge was again evaporated to dryness and re-dissolved in buffer A (20 mM ammonium hydroxide, pH 10) prior to fractionation by high pH reversed-phase chromatography using an Ultimate 3000 liquid chromatography system (Thermo Scientific). In brief, the sample was loaded onto an XBridge BEH C18 Column (130 Å, 3.5 µm, 2.1 mm × 150 mm, Waters, UK) in buffer A and peptides eluted with an increasing gradient of buffer B (20 mM Ammonium Hydroxide in 90% acetonitrile, pH 10) from 0 to 95% over 60 min. 15 fractions were collected by time per sample, evaporated to dryness, and redissolved in 1% formic acid prior to analysis by nano-LC MSMS using an Orbitrap Fusion Lumos mass spectrometer (Thermo Scientific).

*Nano-LC mass spectrometry.* High pH RP fractions were further fractionated using an Ultimate 3000 nano-LC system in line with an Orbitrap Fusion Lumos mass spectrometer (Thermo Scientific). In brief, peptides in 1% (V/V) formic acid were injected onto an Acclaim PepMap C18 nano-trap column (Thermo Scientific). After washing with 0.5% (V/V) acetonitrile 0.1% (V/V) formic acid, peptides were resolved on a 250 mm × 75 µm Acclaim PepMap C18 reverse-phase analytical column (Thermo Scientific) over a 150 min organic gradient, using 7 gradient

segments (1–6% solvent B over 1 min, 6–15% B over 58 min, 15–32%B over 58 min, 32–40%B over 5 min, 40–90%B over 1 min, held at 90%B for 6 min and then reduced to 1%B over 1 min) with a flow rate of 300 nl/min. Solvent A was 0.1% formic acid and Solvent B was aqueous 80% acetonitrile in 0.1% formic acid. Peptides were ionized by nano-electrospray ionization at 2.0 kV using a stainless steel emitter with an internal diameter of 30 µm (Thermo Scientific) and a capillary temperature of 275 °C.

All spectra were acquired using an Orbitrap Fusion Lumos mass spectrometer controlled by Xcalibur 4.1 software (Thermo Scientific) and operated in data-dependent acquisition mode using an SPS-MS3 workflow. FTMS1 spectra were collected at a resolution of 120,000, with automatic gain control (AGC) target of 200,000 and a max injection time of 50 ms. Precursors were filtered with an intensity threshold of 5000, according to charge state (to include charge states 2–7) and with monoisotopic peak determination set to Peptide. Previously interrogated precursors were excluded using a dynamic window (60 s ± 10 ppm). The MS2 precursors were isolated with a quadrupole isolation window of 0.7 m/z. ITMS2 spectra were collected with an AGC target of 10,000, max injection time of 70 ms, and CID collision energy of 35%. For FTMS3 analysis, the Orbitrap was operated at 50,000 resolution with an AGC target of 50,000 and a max injection time of 105 ms. Precursors were fragmented by high energy collision dissociation (HCD) at a normalized collision energy of 60% to ensure maximal TMT reporter ion yield. Synchronous Precursor Selection (SPS) was enabled to include up to 5 MS2 fragment ions in the FTMS3 scan.

*Peptides identification and TMT reporter quantitation.* Acquired raw data files were processed and quantified using Proteome Discoverer software v2.1 (Thermo Scientific) and searched against the UniProt *Mus musculus* database (downloaded November 2018: 81925 sequences) using the SEQUEST algorithm. The raw files from the embryonic and adult samples were processed in two separate batches. For each, peptide precursor mass tolerance was set at 10 ppm, and MS/MS tolerance was set at 0.6 Da. Search criteria included oxidation of methionine (+15.995) and phosphorylation of serine, threonine or tyrosine (+79.966) as variable peptide modifications and carbamidomethylation of cysteine (+57.021) and the addition of the TMT mass tag (+229.163) to peptide N termini and lysine as fixed modifications. Acetylation (+42.011) and Met-loss + Acetylation (−89.030) were included as possible modifications to the protein N-terminus. Searches were performed with full tryptic digestion and a maximum of 2 missed cleavages were allowed. The reverse database search option was enabled and all data was filtered to satisfy the false discovery rate (FDR) of 5%.

*Statistical data analysis.* PSM tables exported from the Proteome discoverer were reprocessed in R using in-house scripts. PSM reporter intensities were scaled by integrated precursor MS1 peak intensities and normalized (Levenberg–Marquardt procedure) across TMT channels, then summed across fractions for each peptidoform (peptide in a specific modification state). The resulting peptidoform expression matrix was then renormalized across samples using VSN normalization (variance stabilization) followed by the Levenberg–Marquardt procedure (alignment of the expression vectors). In addition, the combined embryonic data set was batch-corrected against the litter effect using the ComBat function from the SVA package (batch correction was skipped for the adult data set as PCA analysis revealed only minor litter effects within tissues, and correcting for TMT batch would have removed tissue-specific variation, since each TMT sample was tissue-specific). Peptidoforms were assembled into protein groups, and protein groups-level expression values calculated from those of individual peptidoforms using an in-house weighted average function (using the inverse of each peptidoform's posterior error probability as weights), excluding phosphorylated peptides and their unmodified counterpart form. Average expression values and ratios were calculated for each condition, and an F-test performed using the limma package. For both, embryonic and adult data sets, *P*-value significance thresholds were calculated for a pre-agreed false discovery rate level of 10% (Benjamini–Hochberg procedure). As a second filtering step, regardless of *P*-value, ratios were also not considered if the absolute value of their base-2 logarithm did not exceed a threshold calculated as excluding all but the 5% most extreme ratios between individual controls. For functional annotation clustering, proteins found significant for a particular condition were mapped to GO terms using the DAVID Bioinformatics Resources 6.8 online tool[52,53] (done for proteins at 20% FDR, GO- terms filtered for *P* < 0.01, Benjamini-adjusted *P*-values are reported for significantly enriched terms).

**Western blots.** Littermate embryonic and adult animals (at least *n* = 3 per genotype) were decapitated, the brain was dissected on ice, snap-frozen in liquid nitrogen, and stored at −80 °C until protein extraction. Tissues were homogenized in ice-cold RIPA buffer (50 mM Tris-HCl pH 7.5, 150 mM NaCl, 1% NP40, 0.5% Sodium deoxycholate, 0.1% SDS, 1 mM EDTA, 10 mM NaF) and freshly added protease inhibitors (Roche), lysed for 45 min on ice and centrifuged at 12,000 × *g* for 20 min at 4 °C. Lysates were quantified using the Pierce™ BCA Protein Assay Kit (ThermoFisher, Cat. no. 23225).

For western blots, 25–50 µg of proteins were mixed with 6× Laemmli buffer (375 mM Tris pH = 6.8, 12% SDS, 60% glycerol, 600 mM DTT, 0.06% bromophenol blue), heat-denatured at 95 °C and separated on 8–10% SDS-PAGE

gels in running buffer. Proteins were transferred to a PVDF membrane (Merck) using transfer buffer in a western blotting apparatus (BioRad) for 2 h at 4 °C with 300 mA constant current. The membranes were then blocked with 5% milk in 1× TBST for 1 h at room temperature and incubated with primary antibodies overnight at 4 °C. Secondary anti-IgG antibody coupled to horseradish peroxidase (HRP) was detected using a Pierce™ enhanced chemiluminescent substrate (ThermoFisher) on a GE Healthcare Amersham machine. The following primary antibodies were used: anti-Cul3 (1:800, Cell Signaling #2759), anti-Gapdh (1:1000, Merck ABS16), anti-Pls3 (T-Plastin 1:500, ThermoFisher PA5-27883), anti-Pls3 (T-Plastin 1:1000, Proteintech 12917-1-AP), anti-Pls1 (1:500, Novus Biologicals H00005357-M04), anti-alpha internexin (1:5000, Abcam ab40758), anti-Nischarin B3 (1:200, Santa Cruz sc-365364), anti β-actin (1:5000, Sigma A1978). Secondary antibodies used: donkey anti-rabbit IgG (1:5000, Amersham NA934), goat anti-mouse IgG (1:10,000, Pierce 31432), goat anti-rabbit IgG (1:5000, Goat Dianova 111-035-045) and goat anti-mouse (1:5000, Goat Dianova 111-035-146).

**G-actin/F-actin assay**. The ratio of filamentous (F-actin) and monomer actin (G-actin) was quantified using a G-actin/F-actin in vivo assay kit (Cytoskeleton Inc. cat. nr. BK037) according to manufacturer instructions. Male littermate P21 animals ($n = 3$ per genotype) were decapitated, the brain was dissected on ice, snap-frozen in liquid nitrogen, and stored at −80 °C until assay was performed. Tissues were homogenized in lysis and F-actin stabilization buffer, lysed for 10 min at 37 °C, transferred to a pre-warmed (37 °C) ultracentrifuge (Beckman Coulter, Optima MAX-XP) and spun at $100,000 \times g$ for 1 h at 37 °C to separate the globular actin (supernatant) and the filamentous actin fractions (pellet). The pellets were further re-suspended in depolymerizing buffer and incubated for 1 h on ice to allow actin depolymerization to occur. All samples were diluted with 5× SDS sample buffer, heat-denatured at 95 °C, and separated on 10% SDS-PAGE gels in running buffer. Proteins were transferred to a PVDF membrane (Merck) using transfer buffer in a western blotting apparatus (BioRad) for 2 h at 4 °C with 300 mA constant current. The membranes were then blocked with 5% milk in 1× TBST for 1 h at room temperature and incubated with a primary antibody overnight at 4 °C. A secondary anti-IgG-HRP antibody was detected using a Pierce™ enhanced chemiluminescent substrate (ThermoFisher) on a GE Healthcare Amersham machine. The primary actin antibody was supplied by the kit; secondary antibody used was goat anti-mouse IgG (1:10,000, Pierce 31432).

**RNA isolation and quantitative real-time PCR (qRT-PCR) analysis**. Tissue from E12.5, E14.5, E16.5, P1, P21, and adult C57BL6/J brains was used for wild-type expression analysis. Tissue from $Cul3^{+/-}$ and wild-type E16.5 embryos was used to investigate up-regulated proteins identified through proteomic analyses. For RNA extraction 700 µl Trizol (Invitrogen) and 140 µl chloroform (Sigma) was used for homogenization, followed by centrifugation at $12,000 \times g$ for 15 min at 4 °C. The upper aqueous phase was transferred to a new tube and 1.5 volumes of 100% ethanol (EtOH) were added. RNA was extracted using Zymo-Spin™ IC columns (Zymo Research). In short, the aqueous phase/ethanol mixture was loaded onto the column, washed with 400 µl 70% EtOH before being treated with RQ1 DNAseI (Promega, 5 µl + 5 µl reaction buffer + 40 µl 70% EtOH) for 15 min at RT. After two washes with 70% EtOH the sample was eluted from the column with DEPC-treated H2O. RNA concentration was measured by using the NanoDrop spectro-photometer (Thermo Scientific). 500 ng of RNA were used for cDNA preparation with the RevertAid First Strand cDNA Synthesis Kit (ThermoFisher). cDNA was diluted 1:3 and used for qPCR analysis with Lightcycler 480 Sybr green master mix (Roche) on a Real-Time PCR Roche Lightcycler 480 machine. Samples were run in duplicates or triplicates with the following intron-spanning qPCR primers for target genes: *Cul3 forward* AAGGTGGTGGAGAGGGAACT and *reverse* TCAAACCATTTGGCACACGAC as well as *forward* CAGCGGGTCCTCA CAAAAGA and *reverse* CTGGGTCGGATTCACCTTGT *Pls3 forward* TCTAG AAGGGGAAACTCGGG and *reverse* GGATCACCAGAGCATCCTGC; *Pls1 forward* CCATGCCTACACAAGCCTGA and *reverse* GCGTCTGCAAGGTCA CTGTA; *INA forward* CCAGGCACGTACCATTGAGAT and *reverse* CAATG CTGTCCTGGTAGCCG. As housekeeping genes *Pgk1* (*forward* AAAGTCAGC CATGTGAGCACT and *reverse* ACTTAGGAGCACAGGAACCAAA) and *Gapdh* (*forward* AACGGGAAGCTCACTGGCAT and *reverse* GCTTCACCACCTT CTTGATG) were used. ΔCq expression levels (relative mRNA) were calculated upon normalization to housekeeper genes and plotted.

**Region/cell-type-specific expression of *Cul3***. Brain region-specific RNA-seq data (as RPKM) were downloaded from the BrainSpan Atlas of the Developing Human Brain (http://www.brainspan.org/static/download.html). For the developmental trajectory, prefrontal cortex samples (DFC, MFC, and VFC) were selected and the expression of *CUL3* was plotted for time points up to 1 year of age. For the brain region-specific expression adult samples (starting at 18 years of age) were grouped according to region. After selection, the data was plotted as mean across samples with individual dots representing individual samples.

Cell-type-specific data from scRNA-seq experiments and aggregated by cluster (as log2(CPM + 1)) was downloaded from the Allen Cell Types Database (https://portal.brain-map.org/atlases-and-data/rnaseq, mouse: "Cell Diversity in the Mouse Cortex and Hippocampus", human: "Cell Diversity in the Human Cortex").

Clusters were grouped according to broad cell-type based on the description and hierarchical relationship of the clusters and data was plotted as mean across clusters with individual dots representing individual clusters of the same broad cell type. Some clusters of rare cell types were omitted from the mouse data to simplify the plot.

**Electrophysiology**. Brain slices were obtained from $Cul3^{+/-}$ and wild-type male littermates. Acute coronal slices (300 µm) were prepared from the primary somatosensory cortex. Animals were decapitated under isoflurane anesthesia and whole brains were rapidly removed from the skull and sectioned using a VT 1200 S vibratome (Leica Microsystems) in ice-cold cutting solution, containing (mM): 93 NMDG, 2.5 KCl, 1.2 NaH2PO4, 30 NaHCO3, 20 HEPES, 25 glucose, 5 sodium ascorbate, 2 thiourea, 3 sodium pyruvate, 10 MgCl2, 0.5 CaCl2 (320 mOsm, 7.2–7.4 pH). Slices were recovered at 32 °C for 12 min in the same solution and then allowed to recover at room temperature for at least 1 h in regular artificial cerebrospinal fluid (ACSF), containing (mM): 125 NaCl, 2.5 KCl, 1.25 NaH2PO4, 25 NaHCO3, 25 glucose, 1 MgCl2 and 2 CaCl2 (320 mOsm, 7.2–7.4 pH). The ACSF was continuously oxygenated with 95% O2 and 5% CO2 to maintain the physiological pH. Slices were visualized under infrared-differential interference contrast (IR-DIC) using a BX-51WI microscope (Olympus) with a QIClickTM charge-coupled device camera (Q Imaging Inc, Surrey, BC, Canada).

Patch pipettes (3–5 MΩ; World Precision Instruments) were pulled on a P-1000 puller (Sutter Instruments) and filled with the intracellular recording solution, containing (mM): 115 cesium methanesulphonate, 8 NaCl, 10 HEPES, 0.3 EGTA, 10 Cs4BAPTA, 4 MgATP, 0.3 NaGTP and 0.2% biocytin. Internal pH was adjusted to 7.3 with CsOH and osmolarity adjusted to 295 mOsm with sucrose. Spontaneous postsynaptic currents were recorded from pyramidal neurons in layer 2/3. Excitatory currents (sEPSC) were recorded at holding potential −70 mV, while inhibitory currents (sIPSC) at +10 mV. E/I ratio was evaluated considering only those cells where both sEPSC and sIPSC were recorded. In particular, *E/I* ratio was calculated as the ratio of sEPSC-to-sIPSC amplitudes. Signals were filtered at 2 kHz, digitized at 10 kHz, and acquired using a MultiClamp 700B amplifier and a Digidata 1550 A. Recorded signals were low-pass filtered at 1 kHz and analyzed using Clampfit 10 software (v10.7) (Molecular Devices). Both excitatory and inhibitory synaptic currents were identified by a template created for each neuron using 50–100 single events for each trace. All events recognized through the template were visualized, identified, and accepted by manual analysis. Cumulative distributions for single neurons and recording conditions were obtained by pooling together 300–400 single synaptic currents.

**Generation and culture of neural progenitor cells**. Neural progenitor cells (NPCs) were generated from E13.5 mouse cortices. Briefly, cortical tissues were dissected in L15 media (Sigma cat. nr. L5520). The tissue was dissociated using Accutase (Sigma cat. nr. A6964) for 5 min at 37 °C, pelleted in basal media, resuspended in complete media, and cells plated in uncoated dishes. The next day, cells were dissociated using Accutase, pelleted and re-suspended in complete media, and pooled according to their genotype ($Cul3^{+/+}$ and $Cul3^{+/-}$ NPC) for primary neurosphere formation. Two independent NPC batches were prepared and analyzed. For adherent NPC cultures, dishes were coated with 10 µg/ml Poly-L-ornithine (PLO) (Sigma cat. nr. P3655) and Laminin (3 mg/ml Sigma cat. nr. L2020) and approximately 15,000 cells/cm² cells were plated. $1 \times 10^6$ cells were frozen per freezing vial using a freezing medium containing basal medium, DMSO, and fetal bovine serum (FBS) (8:1:1). Composition of media: Basal medium: 1× DMEM/F12 (Gibco cat. nr. 32500-350), 2 mM Glutamax (Gibco cat. nr. 35050061), 15 mM HEPES (Sigma cat. nr. H0887), 2% D-Glucose (Sigma cat. nr. G879), Sodium bicarbonate (Sigma cat. no. S8761), 100× Penicillin/Streptomycin (Sigma cat. nr. P4333) in sterile water. Complete medium: Basal media plus freshly added 20 ng/ml human recombinant EGF (R&D cat. nr. 236-EG) and 10 ng/ml human recombinant bFGF (R&D cat. nr. 233-FB-025), B27™ Supplement minus VitA (Gibco cat. nr. 12587010).

**In vitro migration assay and live-cell imaging**. For in vitro migration assays, NPCs were dissociated using Accutase for 5 min at 37 °C, pelleted in basal media, and re-suspended in complete media. Two thousand cells were seeded per well in U-bottom 96-well ultralow attachment plates (Corning), assembled into neuro-spheres overnight (14–16 h). Healthy neurospheres (defined by smooth and bright surface) were embedded under glass coverslips in 30 µl of hanging drops of 3D collagen scaffold with a final concentration of 1.7 mg/ml (obtained by mixing bovine collagen (PureCol, Advanced BioMatrix, USA) in 1× minimum essential medium eagle and 0.4% sodium bicarbonate (both Sigma-Aldrich, USA)). Collagen was let to polymerize at 37 °C with 5% CO2 and humidity for 1 h. Next, dishes were flipped around and the second layer of collagen was added on top and let polymerize for another 1 h. Devices were covered with complete media and imaged using a ×10 objective on a brightfield inverted microscope at 37 °C with 5% CO2 for 72 h every 10 min. Calibrated time course TIFF stacks were created in Fiji. Individual cells were tracked manually by an experimenter blinded to the genotype employing the Fiji plugin TrackMate[59]. Cells were chosen randomly from all sites of the sphere at the moment of their detachment and tracked until either re-joining the growing spheres or until the end of the recording. TrackMate data was then

imported into R (3.6.2) environment and analyzed. The track velocity (track displacement/track duration) of detaching cells, their average instantaneous speed (average of all displacements per frame duration), track path length (sum of all displacements between frames), and persistence (track displacement/track path length) were quantified by TrackMate and with R. Finally, results were plotted with the help of ggplot2 (3.2.1), ggpubr (0.2.4) packages.

Alternatively, an adapted neurosphere migration assay was used[60]. Spheres were embedded centrally in 40 µl of 5 mg/ml matrigel (Corning cat. nr. 356234) diluted in ice-cold basal media on PLO/Laminin coated clear bottom 96-well imaging plates (Corning cat. nr. 3603), one sphere per well. After 30 min polymerization of the matrigel at 37 °C, complete media were added to the wells, and images were captured using a ×4 objective on a brightfield inverted microscope, to determine initial sphere sizes. Neurospheres too close to the well-border were excluded from the analysis. 22 and 46 h after embedding, radial outgrowth of NPCs from the sphere was imaged using a ×4 objective on a brightfield inverted microscope and the distance (radius) from the center of the neurospheres was measured in Fiji (v1.52n) and normalized to the initial sphere radius.

**B16-F1 cell culture conditions**. B16-F1 (ATCC® CRL-6323™) mouse melanoma cells were cultivated according to standard tissue culture conditions. Cells were grown at 37 °C in 5% $CO_2$ in DMEM—high glucose (Gibco cat. nr. 11965092), GlutaMAX™ Supplement (Gibco cat. nr. 35050061), Pyruvate (Thermo Scientific) and 10% FBS (#10270-106, Thermo Scientific), 100× Penicillin/Streptomycin (Sigma cat. nr. P4333).

**Generation of Pls3 knockout cell lines using CRISPR/Cas9 and Pls3 over-expression**. Generation of knockout cell lines was performed as previously described[41]. For the generation of Pls3 knockout B16-F1 cell lines (Pls3 KO, Supplementary Fig. 9a–c), CRISPR/Cas9 guide design was performed using http://crispr.mit.edu by choosing a guide sequence targeting exon 3 of Pls3: 5′-TATCGCTAAAAACCTTCCGAA-3′. The following forward and reverse oligo sequences were cloned into a pSpCas9(BB)-2A-Puro (PX459) (Addgene), for stable B16-F1 cell line generation, or a pSpCas9(BB)-2A-GFP (PX458) (Addgene), for transfection of mNPCs, using BbsI-mediated restriction: 5′-CACCGTATCGCT AAAAACCTTCCGAA-3′ and 5′-AAACTTCGGAAGGTTTTAGCGATAC-3′. For guide sequence validation, the genomic DNA sequence flanking exon 3 of the Pls3 target sequence was amplified using Phusion High-Fidelity Polymerase (NEB) from stable Pls3 knockout B16-F1 cell lines (3 different clones #8, #10, #12). Genotyping for the Pls3 knockout allele was performed using the following primers: *forward* ATGCAGCACTGAATGTGTTTGG and *reverse* ACCAATATGTGACCCAAG ACCC. The amplified DNA sequence was cloned into a Zero blunt TOPO vector (Invitrogen) according to manufacturer instructions and sequenced using the following primer: forward CAGGAAACAGCTATGAC. Clones with frameshift mutations causing stop codons downstream of the target site were selected for further characterization (Pls3 KO) and the absence of Pls3 was confirmed via western blot in the Pls3 knockout B16-F1 cell lines (3 different clones #8, #10, #12). The pSpCas9(BB)-2A-GFP plasmid, containing the validated Pls3: 5′-TATCGC-TAAAAACCTTCCGAA-3′ guide sequence was further used for transfections of mNPCs using Lipofectamine™ LTX & PLUS™ Reagent (Invitrogen), cells GFP positive were then analyzed in the random migration assay 24–48 h after transfection.

Transient Pls3 over-expression of Pls3-GFP in NPCs was achieved by the generation of a Pls3-GFP plasmid. To that end, the synthetic full-length Pls3 DNA sequence (mouse) was synthesized by Integrated DNA Technologies (IDT) (gBlocks Gene Fragments) and cloned into Blunt II-TOPO Vector. The DNA sequence from the TOPO vector was then amplified via PCR, introducing Xho- and SalI-restriction sites N- and C-terminally, respectively. The PCR fragment and the EGFP-C1 vector (Clonetech) were digested with the above-mentioned restriction enzymes and ligated. The following primers were used for cloning into EGFP-C1: *Pls3-FW-XhoI-EGFPC1-* ATATCTCGAGATATGGATGAGATGGCGACC and *Pls3-Rev-SalI-EGFPC1-* TTTGTCGACTTACACTCTCTTCATCCC and successful cloning confirmed by sequencing. Pls3 depletion or over-expression was confirmed by qRT-PCR on FACS-sorted cell populations.

**CRISPR-mediated activation of Cul3**. Upregulation of Cul3 levels was performed as previously described using CRISPR-mediated activation[43]. For transcription-mediated activation of Cul3, CRISPR/Cas9 guide design was performed using http://crispor.tefor.net by choosing two guide sequences targeting the promoter of Cul3. The following forward and reverse oligo sequences were cloned into a pAAV-U6-sgRNA-CMV-GFP (Addgene plasmid ID: 85451) using SapI-mediated restriction: 5′-ACCGCGAGACGAGATCCGGCCCA-3′ and 5′-AACTGGGCCGGA TCTCGTCTCGC-3′ as well as 5′-ACCGTCCCCCCTCCACCCTCTGGG-3′ and 5′-AACCCCAGAGGGTGGAGGGGGGAC-3′. Successful ligation of sgRNAs into the vector was confirmed by sequencing using the following primer: forward ACTATCATATGCTTACCGTAAC.

For cell line transfections, two plasmids were used: pAAV-U6-sgRNA-CMV-GFP containing the individual sgRNAs together with pMSCV-LTR-dCas9-VP64-BFP (Addgene plasmid ID: 46912). One day before transfection, 4–5 × 10^5 cells were seeded into individual wells coated with 10 µg/ml Poly-L-ornithine (PLO)

(Sigma cat. nr. P3655) and Laminin (3 mg/ml Sigma cat. nr. L2020) of a 24-well plate. After 24 h incubation in the growth medium, cells were transfected using Lipofectamine™ LTX & PLUS™ Reagent (Invitrogen). In brief, 450 ng of plasmid DNA per well was diluted in 100 µl of Opti-MEM reduced serum medium (Invitrogen) and pre-incubated with 2 µl of PLUS™ reagent. After incubation for 10 min at room temperature, 2 µl Lipofectamine™ LTX was added to the DNA-PLUS™ mixed with Opti-MEM. After an additional incubation for 30 min at room temperature, 100 µl of the DNA-Lipofectamine LTX™-PLUS™ complex was added to each well-containing cells and growth medium. Upregulation of Cul3 transcription was confirmed by qRT-PCR on mixed cell populations. GFP positive cells were then analyzed in the random migration assay 24–48 h after transfection.

**Preparation of NPCs for fluorescence-activated cell sorting (FACS)**. For FACS sorting of GFP+ Cul3+/+ and Cul3+/− transfected cells, 4 days after transfection, NPCs were dissociated using Accutase for 5 min at 37 °C, pelleted in basal media, resuspended in complete media, and kept on ice until sorted. Right before sorting, the cell suspension was passed through a 35 µm cell strainer cap into the test tube (Falcon). FACS was performed on a BD FACS Aria III using a 100 nozzle, sample and 15 ml collection tubes were kept at RT, and NPCs were sorted directly into complete media. Duplet exclusion was performed to ensure the sorting of single cells. Following the sorting of 50,000–200,000 cells, cell suspensions were pelleted, snap-frozen in liquid nitrogen, and stored at −80 °C until RNA or protein extraction.

**Random migration assay**. B16-F1 cells were seeded on #1.5H polymer µ-Slide 4 Well Ph+ coverslips (ibidi GmbH, Germany), which were coated with 30 µg/ml laminin (L-2020; Sigma-Aldrich) in 50 mM Tris, pH 7.4, and 150 mM NaCl buffer for 1 h at room temperature. Brightfield time-lapse images were acquired with a 20x objective on an inverted Nikon Ti2 microscope at 37 °C with 5% $CO_2$ for 15 h every 10 min.

Cul3+/+ and Cul3+/− transfected cells were seeded on #1.5H glass 8 Well chamber coverslips (ibidi GmbH cat. nr. 80841), which were coated with 10 µg/ml Poly-L-ornithine (PLO) (Sigma cat. nr. P3655) for 3 h and Laminin (3 mg/ml Sigma cat. nr. L2020) overnight at 37 °C in 5% $CO_2$. For an in vitro random migration assay, one day after transfection, NPCs were dissociated using Accutase for 5 min at 37 °C, pelleted in basal media, and resuspended in complete media. 7.5 × 10^3 cells were seeded per chamber, and cells attached for 5–6 h before image acquisition.

Brightfield and GFP time-lapse images were acquired with a ×20 objective on an inverted Nikon Ti2 microscope at 37 °C with 5% $CO_2$ for 15 h every 10 min.

**Migration analysis**. Velocity and migration pattern of individual transfected Cul3+/+ and Cul3+/− NPCs were tracked manually employing the built-in Fiji plugin Manual Tracking. If needed, time-lapse images were corrected for drift and adjusted for contrast and brightness using Fiji (v1.50e). Data of traced cells were then imported into the Chemotaxis and Migration Tool (Ibidi, https://ibidi.com/chemotaxis-analysis/171-chemotaxis-and-migration-tool.html), analyzing tracks of individual cells, quantifying the track velocity and cumulative distance.

**STED microscopy**

*Dual-color labeling of neural progenitor cells (NPCs) for actin and tubulin*. NPCs were seeded at a confluency of 70% on #1.5H glass coverslips coated with PLO/Laminin (Marienfeld, Lauda-Königshofen, Germany). After ~12–14 h, they were fixed with 4% paraformaldehyde in PBS for 15 min at room temperature (RT) followed by 3 × 2 min in PBS. Permeabilization was performed with PBS + 0.25% Triton X-100 (Sigma-Aldrich) for 10 min at RT followed by 3 × 2 min in PBS. Cells were blocked with 2% BSA (AppliChem GmbH, Darmstadt, Germany) in PBS for 30 min at RT and then incubated with anti-α-tubulin antibody (T6074, Sigma-Aldrich; 1:1000 in blocking solution) for 1 h at RT. Samples were washed 3 × 3 min in PBS at RT. Incubation with secondary antibody and simultaneously with SiR-actin was done in blocking solution at RT for 1 h (SiR-actin, SC001, Spirochrome, Switzerland; 1:500; goat anti-mouse IgG, conjugated to Alexa Fluor 594, Invitrogen A11005, 1:500). Samples were washed 3 × 3 min in PBS at RT and then incubated with DAPI (D9542, Sigma-Aldrich; 1:5000) in PBS at RT for 5 min. Samples were washed 3 × 2 min in PBS, then mounted with Dako Fluorescence Mounting Medium (S3023, Agilent Technologies).

*STED microscopy*. STED imaging was performed on an Abberior Instruments Expert Line STED microscope. STED wavelength was 775 nm and excitation was performed at ~560 nm and ~640 nm. A ×100/1.4 NA oil immersion objective (Olympus, UPLSAPO 100XO) was used. STED pulses had a duration of ~1 ns and time gating was applied throughout. The resolution was increased in the *xy*-direction with a "doughnut" shaped beam and in the z-direction with an additional z-STED pattern according to the power ratio given below. Power levels are given as power at the back aperture of the objective lens with an estimated uncertainty of ~15%.

*Imaging parameters*. Figure 7b left Cul3+/+: STED: pixel size 30 nm × 30 nm; pinhole size 1 airy unit; excitation laser powers: 560 nm: 2.3 µW; 640: 6.2 µW; STED: 90 µW with lateral/axial power ratio 75/25. Pixel dwell time: 15 µs with 2 scans per line for 560 nm excitation and 4 scans per line for 640 nm excitation.

*Figure 7b right Cul3*$^{+/-}$: STED: pixel size 30 nm × 30 nm; pinhole size 1 airy unit; excitation laser powers: 560 nm: 2.3 μW; 640: 6.2 μW; STED: 90 μW with lateral/axial power ratio 75/25. Pixel dwell time: 15 μs with 2 scans per line for 560 nm excitation and 4 scans per line for 640 nm excitation.

*Figure 7e left Cul3*$^{+/+}$: STED: pixel size 30 nm × 30 nm; pinhole size 1 airy unit; excitation laser powers: 560 nm: 2.3 μW; 640: 6.2 μW; STED: 90 μW with lateral/axial power ratio 75/25. Pixel dwell time: 15 μs with 2 scans per line for 560 nm excitation and 4 scans per line for 640 nm excitation.

*Figure 7e right Cul3*$^{+/-}$: STED: pixel size 30 nm × 30 nm; pinhole size 1 airy unit; excitation laser powers: 560 nm: 2.3 μW; 640: 6.2 μW; STED: 90 μW with lateral/axial power ratio 75/25. Pixel dwell time: 15 μs with 2 scans per line for 560 nm excitation and 4 scans per line for 640 nm excitation.

*Figure 7h left Cul3*$^{+/-}$: STED: pixel size 30 nm × 30 nm; pinhole size 1 airy unit; excitation laser powers: 560 nm: 2.3 μW; 640: 6.2 μW; STED: 90 μW with lateral/axial power ratio 75/25. Pixel dwell time: 15 μs with 2 scans per line for 560 nm excitation and 4 scans per line for 640 nm excitation.

*Figure 7h right Cul3*$^{+/+}$ *Pls3 over-expression*: STED: pixel size 30 nm × 30 nm; pinhole size 1 airy unit; excitation laser powers: 560 nm: 3.7 μW; 640: 6.2 μW; STED: 146 μW with lateral/axial power ratio 75/25. Pixel dwell time: 15 μs with 2 scans per line for 560 nm excitation and 3 scans per line for 640 nm excitation.

*Figure 7k left Cul3*$^{+/-}$: STED: pixel size 30 nm × 30 nm; pinhole size 1 airy unit; excitation laser powers: 560 nm: 2.3 μW; 640: 6.2 μW; STED: 90 μW with lateral/axial power ratio 75/25. Pixel dwell time: 15 μs with 2 scans per line for 560 nm excitation and 4 scans per line for 640 nm excitation.

*Figure 7k right Cul3*$^{+/+}$ *Pls3-over-expression*: STED: pixel size 30 nm × 30 nm; pinhole size 1 airy unit; excitation laser powers: 560 nm: 3.7 μW; 640: 6.2 μW; STED: 146 μW with lateral/axial power ratio 75/25. Pixel dwell time: 15 μs with 2 scans per line for 560 nm excitation and 3 scans per line for 640 nm excitation.

**Orientation analysis**. The orientation of actin and tubulin fibers was analyzed in custom-written Python routines using the scikit-image library[61] (ver. 0.16.2). We extracted the orientation distributions based on the structure tensor. The structure tensor summarizes local orientations and their coherence (degree of anisotropy) at each image location[62]. For each image of size 30 μm × 30 μm with pixel size 30 nm × 30 nm (1000 pixel²), we first normalized the raw photon counts from 16-bit integers to values in the range [0, 1] by subtracting the minimum gray value, followed by dividing with the maximum. From the structure tensor computed at Gaussian scale $\sigma = 60$ nm (2 pixel), we extracted local orientations, coherency, and energy (i.e., image gradient magnitude). Coherence yields values of 1 when the local structure is totally aligned, and 0 when there is no preferred direction. The energy measures the magnitude of the local structure (gradient). The resulting orientations were visualized in the hue, saturation, and brightness (HSB) color-space, where we set hue as the orientation angle, saturation as coherency, and brightness the normalized gray-value of the source image. The distribution of orientations was built as the histogram of orientations weighted by the corresponding coherency. Image locations with coherence or energy <10% of their maximum value were excluded from the histogram. Each orientation histogram was aligned to the corresponding dominant orientation angle computed from the tubulin channel. The dominant orientation is computed as the average orientation angle inside the cell. Each cell was segmented manually prior to analysis.

After alignment, we averaged all orientation histograms per group (*Cul3*$^{+/+}$, n = 43; *Cul3*$^{+/-}$, n = 43, from three independent NPC preparations respectively) and computed 95% confidence intervals per orientation angle. To test statistical significance, we measured the spread of each orientation distribution as the standard deviation of angles weighted by their occurrences. Reported P-values were computed using a two-tailed Welch's t-test using the scipy library (ver.1.3.0).

**Adhesion site analysis**. Adhesion sites in the actin channel were analyzed in custom-written Python routines using the scikit-image library. Each cell was segmented manually prior to analysis. For each image of size 30 μm × 30 μm with pixel size 30 nm × 30 nm (1000 pixel²), we first normalized the raw photon counts from 16-bit integers to values in the range [0, 1] by subtracting the minimum gray value, followed by dividing with the 99.9$^{th}$ percentile. We counted the number of adhesion sites per cell as the number of local maxima in the scale-space of a series of Laplacian of Gaussian (LoG) filters (https://doi.org/10.1007/BF01469346). A LoG filter with scale $\sigma$ is a high-pass filter with a strong response at bright, blob-like structures of radius $\sqrt{2}\sigma$. We computed a series of 5 LoG filters for $\sigma = 3, 4, 5, 6,$ and 7 pixel (90, 120, 150, 180, and 210 nm). A local maximum was considered an adhesion site if it exceeded a threshold of 0.25 in the LoG filter response. To exclude detection at the cell boundary we removed adhesion sites closer than 1 μm from the manually segmented cell boundary. To compute the density of adhesion sites, we divided the number of detected adhesion sites by the area of the cell. To test statistical significance we used a two-tailed Welch's t-test using the scipy library (ver 1.3.0).

**Tamoxifen-induced *Cul3* deletion**. To induce *Cul3* deletion in adult double transgenic mice, *Cul3*$^{+/fl}$ *Cag*-CreER (Cag-CreER line: Jackson 04453), were injected intraperitoneally with Tamoxifen (Sigma T5648) (100 mg/kg body weight, 10 mg/ml stock solution in corn oil) or vehicle (corn oil). Animals, aged P30–P40, were injected for five consecutive days and behavior tests were performed >21 days post (last)

injection (animals age P55-P65) to ensure successful recombination, protein degradation, and Tamoxifen (and its metabolites) clearance, as previously described[63]. Behavioral tests were performed as described above. After behavioral tests, animals were sacrificed, the brain dissected and the right hemisphere was used for tissue lysis and western blot analysis to determine Cul3 protein levels and success of Tamoxifen-induced deletion. As described previously[64], in 5–20% of mice treated with Tamoxifen, induction appeared to fail altogether, and no changes on Cul3 protein levels could be observed. These mice were excluded from the analysis.

**Statistics**. Statistical analyses were performed using Microsoft ® Excel ® 2013, Origin Software (Origin Inc.) and GraphPad Prism 6/8. Shapiro–Wilk test was used to evaluate normal distribution, means and standard deviations of the data. Parametric data were analyzed for significance using unpaired two-tailed t-tests, 1-way or 2-way ANOVAs with Sidak's post-hoc test, using *P < 0.05, **P < 0.01, and ***P < 0.001 for significance, and presented as a bar, box and whiskers, scatter dot plots and mean ± standard error of the mean (SEM), unless otherwise specified. Data sets with non-normal distributions were analyzed using the two-tailed Mann–Whitney U test. Adjustment for multiple comparisons was made using post-hoc tests. Cumulative probability plots of the amplitude and inter-event interval of synaptic currents were compared with the Kolmogorov–Smirnov two-sample test. Illustrations for figures were prepared in Adobe Illustrator or using BioRender (BioRender.com). Detailed statistics are presented in Supplementary Data 1. Experiments were replicated at least three times.

**Reporting summary**. Further information on research design is available in the Nature Research Reporting Summary linked to this article.

## Data availability
All data supporting the findings of this study are available within the article and its supplementary information files or from the corresponding author upon reasonable request. The mass spectrometry proteomics data have been deposited to the ProteomeXchange Consortium via the PRIDE[65] partner repository with the data set identifier PXD017040. Publicly available data sets used are available from the Allen Cell Types Database [https://portal.brain-map.org/atlases-and-data/rnaseq] and the BrainSpan Atlas [http://www.brainspan.org/static/download.html]. Source data are provided with this paper.

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

## Acknowledgements

We thank A. Coll Manzano, F. Freeman, M. Ladron de Guevara, and A. Ç. Yahya for technical assistance, S. Deixler, A. Lepold, and A. Schlerka for the management of our animal colony, as well as M. Schunn and the Preclinical Facility team for technical assistance. We thank K. Heesom and her team at the University of Bristol Proteomics Facility for the proteomics sample preparation, data generation, and analysis support. We thank Y. B. Simon for kindly providing the plasmid for lentiviral labeling. Further, we thank M. Sixt for his advice regarding cell migration and the fruitful discussions. This work was supported by the ISTPlus postdoctoral fellowship (Grant Agreement No. 754411) to B.B., by the European Union's Horizon 2020 research and innovation program (ERC) grant 715508 (REVERSEAUTISM), and by the Austrian Science Fund (FWF) to G.N. (DK W1232-B24 and SFB F7807-B) and to J.G.D (I3600-B27).

## Author contributions

J.M. and L.A.S. designed and performed experiments, analyzed data, and prepared figures. B.B., G.D., and S.T. performed experiments and data analysis. A.N., C.S., and C.P.D. analyzed data. C.K., Z.D., L.S.K., and E.C. performed experiments. J.G.D. supervised STED-imaging. F.K.S. supervised generation and analysis of Pls3 mutant cells. G.N. conceived and supervised the study. G.N. wrote the paper together with J.M. and L.A.S. All authors read and approved the final version of the manuscript.

## Competing interests

The authors declare no competing interests.
