## [Peer Review File · Nature Communications]

Reviewer #1 (Remarks to the Author):

Morandell, Schwarz, Novarino and colleagues present an interesting manuscript investigating the molecular and cellular mechanisms underlining Cul3 haploinsufficiency. The authors report that Cul3 haploinsufficiency plays a crucial role during development in establishing ASD-associated behavioral phenotypes, and links this to a role of Cul3 in neuronal migration in cortex. The authors then performed TMT-proteomics and found that Cul3 regulates the expression of cytoskeletal and adhesion proteins. They focus on Cul3s regulation of the protein Pls3, and find a novel role for Pls3 in regulating neuronal migration. Finally, the authors use a CRISPR-mediated approach to restore Cul3 expression in neuronal cells to restore the cell migration phenotype.

The paper is well-written and mostly easy to follow. The findings are interesting and novel, and I believe they will be of interest to the field. The main strength of the paper is the demonstration of a crucial developmental role for Cul3 and the important information on the mechanisms underlying the Cul3 haploinsufficiency phenotypes. Importantly, the paper uses the more disease-relevant Cul3^{+/-} mice in all experiments instead of reliance on only Cre-lines, which may increase the translatability of the findings. The data appear of high quality, with a convincing number of repeats and statistical analyses. I have some concerns and a few comments that I hope can improve the manuscript before publication.

- 1) For Figure 1 and associated supplementary figures, the split of sexes for some experiments but not for others makes the data needlessly confusing and difficult to follow, especially since any mention of the sex of the animals are absent for all other experiments. I applauded the authors decision to investigate sex differences, but the way the data is presented now makes it unclear if the significant differences really is present for both sexes, only in one sex, or only in the pooled data because of an increase of animals. For example, in figure 1f, the social novelty index is only one star significant, thus raising the question if this phenotype is significant if you would split it by sexes. If the conclusion is that sex differences are not prevalent, the author should focus on the pooled data in Figure 1 and show the few existing sex differences in the supplementary.
- 2) For the behavioral experiments in Figure 2, the authors should state if this is done using both sexes, or if it is split by sexes in the same way as Figure 1. Because some of the behaviors tested only had a significant finding in one sex, the use of a mixed sex cohort in the tamoxifen-induced mice makes the data harder to interpret. See for example the results in Figure 2d' vs 3e'.
- 3) The authors performed a global proteomic analysis of forebrain tissue from Cul3^{+/-}, Cul3^{+/fl} Emx1-Cre and Cul3^{fl/fl} Emx1-Cre mice. However, the overlap between genotypes is very low, especially between the Cul3^{+/-} and the Cul3^{+/fl} Emx1-Cre mice. Since the cortical layer phenotype in these animals are very similar, and the argument is that the proteomic analysis gives mechanistical insight, this low overlap makes the data more difficult to trust. One way to give more confidence in the data is to include a comparison of the ratios instead of P-values between the different genotypes, since P-values are more variable between experiments then ratios. If the direction of change is similar for most proteins, this would give some confidence that the changes in the proteomes between the genotypes are more comparable and biologically meaningful.
- 4) Knocking out or overexpressing Pls3 in neural progenitor cells affect the speed of cell migration (Figure 7a-c). However, there is no data showing the level of knockdown achieved in primary neural cells, and no data at all on the level of overexpression achieved. At minimum, the authors should show the level of overexpression on the protein level to be able to correctly interpret the data, especially since the effect of overexpression is much stronger then what should be a complete knockout.
- 5) Similarly, in 7d-f, knockout of Pls3 rescues the mobility phenotype of Cul3^{+/-} cells. The authors do not show if the level of Pls3 is increased in Cul3^{+/-} cells, or the level of Pls3 after the knockout. It would be somewhat surprising if complete Pls3 knockout (as reached in supplementary figure 8a) rescues migration to baseline in Cul3^{+/-} cells, since Pls3 knockout has a phenotype of its own. Furthermore, it is unclear how Cul3 can regulate migration via Pls3 if removal of Pls3 restores the phenotype.

6) The authors use a CRISPR-based approach to rescue Cul3 expression in primary neuronal cultures. A question not addressed is if this CRISPR-based rescue also restores Pls3 levels. This would more closely tie the two mechanisms together.

Minor comments:

1) Several observations in this paper, including behavioral, electrophysiological, and biochemical analysis, are different and in some cases completely opposite from a recent paper in Neuron from Dong et al., (2020). This is only briefly mentioned in the text, and largely explained by differences in Cre-lines used. Although it is true that the differences in Cre-lines used could explain this, the large differences in phenotypes (for example I/E balance) is still interesting and deserves a more thorough discussion.

2) The abstract's final statement on evidence for a therapeutic approach is too strong for an in vitro rescue of an in vitro phenotype.

3) Cux3 heterozygous mutation results in a reduction of PV cells by around 25% (Fig. 4f), but the relative count of PV cells is the same for all layers. This seems difficult to reconcile. Could the authors elaborate on this?

4) Of note, in the supplementary figure 2d the dot plots for total distance and velocity for Cul3+/- mice look completely identical. The author should make sure that this is not a mistake in the graph.

Reviewer #2 (Remarks to the Author):

In this manuscript, the authors investigate the role of one of the high-risk ASD genes that encode for Cul3 in cortical development. By performing a battery of behavioural tests in the constitutive heterozygous Cul3 KO mouse, the authors revealed motor deficits accompanied by aberrant social novelty exploration. Interestingly, deletion of Cul3 in adulthood only impaired rotarod task. The authors found that Cul3 heterozygous mutation results in deficits in neural cell migration and cortical lamination as well as changes in spontaneous frequency both of excitatory and inhibitory transmission. No reduction in the dendritic length, branching or in the number of dendrites and spines were revealed. The authors identified up and down regulated proteins in the Cul3+/- and Cul3 +/- Emx1-Cre embryonic cortexes. Among the several proteins, the analysis pointed to proteins involved in the cytoskeletal organization and in particular to PIs3. By increasing Cul3 expression, the authors were able finally to rescue the neural cell migration defects.

The manuscript is interesting and well written, the approaches are state of the art and the quality of the figures is excellent. Several points and limitations are well discussed in the discussion and I think that overall the paper offers important elements for possible drug development.

I have some comments:

The rationale underlying the choice of investigating somatosensory cortex is not clear. Indeed, considering the behavioural and anatomical data, I would have expected that the authors would have focused their study on the cerebellum. Do the Emx1-Cre mice also show behavioural deficits?

In the behavioural experiments, it is not clear why the authors only separated between male and female for the motor task and not for the other experiments.

The electrophysiological data are not convincing. Indeed, it seems that Cul3 downregulation affects the inhibitory and excitatory spontaneous frequency more than the amplitude. How the E/I ratio was calculated? I would encourage to avoid the term as slightly shifted or to write "GABAergic transmission is less affected". Indeed, the comparison has not been done statistically.

Although the authors do not find changes in the number of dendrites and spines, they should

investigate whether the spine morphology is affected. Along this line, the author should as well investigate the G actin/F actin ratio to have more information regarding the role of actin polymerization.

Are PI3 level normalized by enhancing Cul3 gene transcription?

Minor:

In Figure 1d, although the data indicate that mutant mice do not reach the same level of motor performance as the control by the third day, male mice show an increase in their performance. I would suggest that the expression, motor learning impairment is too strong and not supported by the data.

Reviewer #3 (Remarks to the Author):

This study provides a thorough description of behavioral alterations and cortical malformations arising in an haploinsufficient Cul3 mouse model. The behavioral analysis is impressive in its rigor and reveals locomotor defects, lack of interest in social novelty and alterations in cognition. This work is also remarkable by its multidisciplinary, ranging from the analysis of mouse behavior, histology and physiology to proteomic screening. The manuscript is clearly written, the behavioral analysis is very impressive, but in its present form, the molecular analysis falls short of presenting a clear disease mechanism. Additional experiments attempting to better understand the cause of the cortical malformations and whether those contribute to the behavioral phenotype would dramatically increase the impact of the manuscript. Some key experiments should thus be redesigned to more directly address the claims of the manuscript.

Major issues

About the behavioral analysis

> The social novelty phenotype is sometimes referred in the manuscript to as "reduced social memory" or "interest in social novelty". In my opinion the latter is a better way to phrase it. Indeed, the behavioral analysis does not convincingly show that lack of interest in social novelty is due to memory alterations since memory retention is not impaired. In addition, interest in social novelty can also be reduced following impairment in the reward circuitry, independently of memory alterations. Additional experiments addressing the implication of memory on the social phenotype could be carried out, for instance using the object recognition test.

About the reduction of cortical thickness.

>It is unclear how the reduction of cortical thickness occurs in Cul3 het animals. Migration defects are unlikely to contribute to this phenotype, thus proper analysis of cell cycle progression and cell cycle exit should be performed. Moreover, when looking at embryonic Cul3cKO, the thickness reduction is mostly affecting upper layers, pointing to a possible IP defect. This should be addressed and the expression pattern of Cul3 should also be carefully analyzed in the developing cortex. Reduction of cortical thickness can also arise from change of cell density, this should be looked at.

About the migration defects.

> Analyzing the position BrdU+ cells in the cortex is not sufficient to take strong conclusion on migration and neurosphere assays are not very useful here, as the identity of migrating cell is unknown (they are also moving too fast as to be bona fide projection neurons). The analysis of the migration defects and their rescue should be done by time-lapse recording on cultured brain slices.
>The reduction of GABAergic interneurons described in the mutant is not supporting a migration defect, as it could arise from the poor production/specification or even cell survival when Cul3 is not properly expressed
>The in vitro data provided with Sir-Actin are interesting but difficult to link to the actual projection neurons defect that harbor a very stereotyped shape during migration, which is not ameboid.

About molecular mechanism.

How is reduction of Cul3 leading to increase of Pls3? Is it direct, can the authors provide some data to bridge the gap?

Minors

Line 51: Indirectly was meant to be inversely ?

Discuss gender differences in stance length

There are no statistical differences in cerebellar area, thus the sentence 147-148 is not correct

Line 212: no demonstration has been given to support that the migration defects is cell intrinsic

Line 231: can you explain how the E/I ratio has been obtained?

Reviewer #4 (Remarks to the Author):

Morandell and colleagues investigate the function of the autism-associated Cul3 gene, a ubiquitin ligase, and perform an extensive phenotypic analysis of Cul3 mutant mice. The authors show behavioral abnormalities related to social memory and cognition. They observe cell migration defects in cortex in Cul3 mutants and identify a number of cytoskeleton-associated proteins as potential targets of Cul3 using proteomics. They show that the Pls3 protein is upregulated in Cul3 mutant tissue and that reducing Pls3 levels in Cul3 mutant neural progenitor cells (NPCs) normalizes abnormal migratory behavior of these cells.

The paper uses various Cul3 mouse lines and contains many different types of data, some of which interrupt the flow of the study and have little relevance to the rest of the study (e.g. electrophysiological and morphological analysis in Figure 5); at the same time related data is sometimes spread out over multiple figures (e.g. Figures 3 and 4). Some conclusions are overstated. The design of key experiments (proteomic analysis in Figure 6) that lay the mechanistic basis of the study is very difficult to assess. This makes the paper difficult to read in places. A condensed version focusing on the key finding of the role of Cul3 in cell migration would make this manuscript more accessible.

Major Points:

1. Figure 2: the authors test the behavioral effects of heterozygous deletion in late postnatal animals using tamoxifen-injected Cul3^{fl/+} Cag-CreER mice. These animals do not show major behavioral abnormalities, in contrast to constitutive Cul3^{+/-} heterozygous mice (Figure 1), leading the authors to conclude that early developmental function of Cul3 is critical for the appearance of behavioral abnormalities later in life. It should be noted that the lines used in these experiments have a different genetic background. Early-stage Cul3 deletion using tamoxifen-injected Cul3^{fl/+} Cag-CreER mice would have allowed for a better comparison in animals of the same genetic background.

2. Figure 6: the authors perform proteomic analysis on multiple brain regions of different genotypes at various ages using TMT-labelling. Because of the complexity of their experiments, it should be much clearer which conditions have actually been run in the same TMT experiment. The current description in the methods section is unclear and leaves the reader guessing. This is especially the case for the embryonic Emx1-cre samples, where n=3/genotype as stated in the Methods does not explain how the experiment is set up within a TMT 11-plex. The design should be clearly outlined in the Methods. A cartoon and description in the figure legend outlining the different conditions of the TMT experiment would help to make the design clear without consulting the Methods.

3. Figure 6: identification of Pls3 and Ina as potential Cul3 targets. The authors convincingly show that the levels of Pls3 and Ina are altered in Cul3 mutant tissue, but there is no direct evidence

that these proteins are ubiquitylated by Cul3. This should be experimentally demonstrated.

4. Figure 6: the False Discovery Rate of 20% that the authors use to determine up- and down-regulated proteins is very lenient compared to common standards. This is especially obvious in Fig 6 panels c and d where multiple protein groups in the scattered "base" (high ratio/low p-value; meaning high variability) of the volcano plots are considered significant. These would be filtered out when a more stringent FDR is used, as is actually shown by the authors in the supplementary tables S3-5,8 that include a more acceptable 10% FDR. The impression is raised that this leniency is only used so that GO term enrichment analysis can be applied and an overstated link to cytoskeleton organization can be made, while their main targets of interest (Pls3 and INA) are actually very significantly upregulated and would survive harsher filtering. The association between Cul3 mutations and cytoskeletal-associated protein regulation in general is overstated again in the adult dataset where only Vimentin and GFAP are identified just once without any wider cytoskeletal effect.

The authors should accept that their datasets do not show major changes in the proteome of their Cul3 mice and not claim overblown conclusions based on too soft filtering. Instead they should keep their focus on the targets that are actually significantly regulated and validate them like Pls3 and INA. In that regard it is interesting that the authors chose to completely ignore Wdfy1, which is reproducibly upregulated in all their datasets but is not a cytoskeletal-associated protein.

5. Figure 7: in the text and figure title, the authors imply that the disorganized actin cytoskeleton observed in Cul3^{+/-} cells is due to the altered levels of Pls3. To demonstrate this the authors should investigate the actin cytoskeleton in NPCs with altered Pls3 levels (KO and overexpression).

Minor Points:

1. Figure 6e: GO analysis: please clarify the origin of the numbers of proteins (178 up, 140 down) used in this analysis: were these significantly regulated proteins or was this analysis done on all regulated proteins, regardless of whether they are significant or not?

2. Figure 6i: Was Nisch identified as a differentially regulated protein in Cul3^{+/-} mice as well? Please clarify.

3. Images in 7g-l are difficult to interpret. Please provide clearer examples that show the actin cytoskeleton at the cell's edge.

4. The flow of the manuscript could be improved by focusing on the key findings and moving Figure 2, 4f,g, and Figure 5 to the supplement; Figure 3 and 4 could be combined.

RESPONSE

Reviewer #1 (Remarks to the Author):

Morandell, Schwarz, Novarino and colleagues present an interesting manuscript investigating the molecular and cellular mechanisms underlining Cul3 haploinsufficiency. The authors report that Cul3 haploinsufficiency plays a crucial role during development in establishing ASD-associated behavioral phenotypes, and links this to a role of Cul3 in neuronal migration in cortex. The authors then performed TMT-proteomics and found that Cul3 regulates the expression of cytoskeletal and adhesion proteins. They focus on Cul3s regulation of the protein PIs3, and find a novel role for PIs3 in regulating neuronal migration. Finally, the authors use a CRISPR-mediated approach to restore Cul3 expression in neuronal cells to restore the cell migration phenotype.

The paper is well-written and mostly easy to follow. The findings are interesting and novel, and I believe they will be of interest to the field. The main strength of the paper is the demonstration of a crucial developmental role for Cul3 and the important information on the mechanisms underlying the Cul3 haploinsufficiency phenotypes. Importantly, the paper uses the more disease-relevant Cul3^{+/-} mice in all experiments instead of reliance on only Cre-lines, which may increase the translatability of the findings. The data appear of high quality, with a convincing number of repeats and statistical analyses. I have some concerns and a few comments that I hope can improve the manuscript before publication.

We thank the reviewer for the positive feedback, the valuable suggestions and detailed comments. Please find a point-by-point response below.

1) For Figure 1 and associated supplementary figures, the split of sexes for some experiments but not for others makes the data needlessly confusing and difficult to follow, especially since any mention of the sex of the animals are absent for all other experiments. I applauded the authors decision to investigate sex differences, but the way the data is presented now makes it unclear if the significant differences really is present for both sexes, only in one sex, or only in the pooled data because of an increase of animals. For example, in figure 1f, the social novelty index is only one star significant, thus raising the question if this phenotype is significant if you would split it by sexes. If the conclusion is that sex differences are not prevalent, the author should focus on the pooled data in Figure 1 and show the few existing sex differences in the supplementary.

Response: We thank the reviewer for this observation and agree that presenting both mixed and split datasets for males and females made it difficult to follow the main message of the behavioral findings. As suggested by the reviewer, we now present pooled datasets for all the behavioral tests in the main figure (Fig. 1) and show the few existing sex differences in the Supplementary Fig. 2b (e.g. initial coordination on the RotaRod).

2) For the behavioral experiments in Figure 2, the authors should state if this is done using both sexes, or if it is split by sexes in the same way as Figure 1. Because some of the behaviors tested only had a significant finding in one sex, the use of a mixed sex cohort in the tamoxifen-induced mice makes the data harder to interpret. See for example the results in Figure 2d' vs 3e'.

Response: Since sex differences appear to be minor in *Cul3^{+/-}* mice (see above), for the tamoxifen injections we analyzed both males and females together. For Fig. 2 the data from males and females were pooled (like in Fig. 1). Additionally, to be consistent with the way we presented the data for *Cul3^{+/-}* mice, in the revised version of the manuscript we present the data of the initial coordination on the RotaRod split by sex also for the tamoxifen-injected cohort (Supplementary Figure 2h).

3) The authors performed a global proteomic analysis of forebrain tissue from *Cul3^{+/-}*, *Cul3^{+/fl}* *Emx1-Cre* and *Cul3^{fl/fl}* *Emx1-Cre* mice. However, the overlap between genotypes is very low, especially between the *Cul3^{+/-}* and the *Cul3^{+/fl}* *Emx1-Cre* mice. Since the cortical layer phenotype in these animals are very similar, and the argument is that the proteomic analysis gives mechanistical insight, this low overlap makes the data more difficult to trust. One way to give more confidence in the data is to include a comparison of the ratios instead of P-values between the different genotypes, since P-values are more variable between experiments than ratios. If the direction of change is similar for most proteins, this would give some confidence that the changes in the proteomes between the genotypes are more comparable and biologically meaningful.

Response: We thank the reviewer for this suggestion. In the revised manuscript we provide a comparison of the ratios at log 0.1 fold change instead of P-values for the embryonic datasets (current Fig. 5 f, g). Indeed, using ratios rather than P-values, we observe that the fraction of proteins with similar changes in the different mutants is much higher than what we reported in the previous version of the manuscript.

4) Knocking out or overexpressing *Pls3* in neural progenitor cells affect the speed of cell migration (Figure 7a-c). However, there is no data showing the level of knockdown achieved in primary neural cells, and no data at all on the level of overexpression achieved. At minimum, the authors should show the level of overexpression on the protein level to be able to correctly interpret the data, especially since the effect of overexpression is much stronger than what

should be a complete knockout.

Response: We agree with the Reviewer that showing Pls3 expression levels is critical for the interpretation of the data presented in Fig. 7a-c (now Fig. 6a-c). Since for the Pls3 overexpression and CRISPR-based loss of function experiments the transfection efficiency was too low to obtain enough cells for WB analysis, to evaluate *Pls3* expression levels, we performed quantitative PCR analysis of FACS-sorted GFP positive NPCs. QPCR analysis revealed a 50% increase and 60% reduction in *Pls3* mRNA levels in NPCs with Pls3 overexpression or loss of function, respectively (Supplementary Fig. 9d). The fact that we did not see a 100% reduction in *Pls3* mRNA levels in the loss of function experiment is compatible with the fact that for the NPC experiment we employed a transient expression system and assayed *Pls3* expression levels (and migration) 48 hours after transfection with the CRISPR-Cas9 system.

5) Similarly, in 7d-f, knockout of Pls3 rescues the mobility phenotype of *Cul3*^{+/-} cells. The authors do not show if the level of Pls3 is increased in *Cul3*^{+/-} cells, or the level of Pls3 after the knockout. It would be somewhat surprising if complete Pls3 knockout (as reached in supplementary figure 8a) rescues migration to baseline in *Cul3*^{+/-} cells, since Pls3 knockout has a phenotype of its own. Furthermore, it is unclear how *Cul3* can regulate migration via Pls3 if removal of Pls3 restores the phenotype.

Response: We agree that this information is crucial. In the revised version of the study, we included a WB showing increased levels of Pls3 protein also in *Cul3*^{+/-} NPCs (Supplementary Fig. 9f). Additionally, as discussed above, we confirmed that transfection of the CRISPR-Cas9 with the *Pls3* single guide RNA leads to a significant reduction of *Pls3* mRNA levels in FACS sorted GFP+ cells (Supplementary Fig. 9d). While, as expected, the reduction of *Pls3* mRNA levels is exactly the same in wild-type and *Cul3*^{+/-} mutant cells (Supplementary Fig. 9d, purple bars), under our experimental conditions the *Pls3* mRNA reduction does not reach 100% (see comment above). Since we assume that the Pls3 left over in *Cul3*^{+/-} cells cannot be properly degraded, we expect Pls3 protein levels to be higher in the *Cul3*^{+/-} background than in wild-type cells. This explains why under our experimental conditions we do observe a full rescue of the migration defect rather than a defect in the opposite direction, like in the *Pls3* KO cells.

6) The authors use a CRISPR-based approach to rescue *Cul3* expression in primary neuronal cultures. A question not addressed is if this CRISPR-based rescue also restores Pls3 levels. This would more closely tie the two mechanisms together.

Response: We thank the reviewer for this suggestion. We now quantified Pls3 protein levels in CRISPR-rescued *Cul3*^{+/-} NPCs. As expected, we found that increasing *Cul3* levels is sufficient to

reduce Pls3 levels in cells (Supplementary Fig. 11b), further supporting a regulatory role of Cul3 for Pls3. We would like to add that when employing the CRISPR-Cas9 activator system we are able to obtain a much higher transfection efficiency than when transfecting cells with the CRISPR-Cas9-loss of function system. Therefore, in this case, we were able to obtain a sufficient quantity of GFP+ cells to quantify Pls3 protein levels by WB.

Minor comments:

1) Several observations in this paper, including behavioral, electrophysiological, and biochemical analysis, are different and in some cases completely opposite from a recent paper in Neuron from Dong et al., (2020). This is only briefly mentioned in the text, and largely explained by differences in Cre-lines used. Although it is true that the differences in Cre-lines used could explain this, the large differences in phenotypes (for example I/E balance) is still interesting and deserves a more thoroughly discussion.

Response: We agree with the Reviewer and have included a more thorough discussion of these discrepancies in the discussion of the revised manuscript.

2) The abstracts final statement on evidence for a therapeutic approach is too strong for an in vitro rescue of an in vitro phenotype.

Response: We thank the Reviewer for the observation and rephrased this statement accordingly.

3) Cux3 heterozygous mutation results in a reduction of PV cells by around 25% (Fig. 4f), but the relative count of PV cells is the same for all layers. This seem difficult to reconcile. Could the authors elaborate on this?

Response: We want to clarify that in the right panel of Fig. 4 f (current Supplementary Fig. 6d) the number of PV+ cells per layer was normalized to the total number of PV+ cells in the respective cortical column. The fact, that in the layer distribution no single layer seems to be exclusively affected by a loss of PV+ cells indicates that the reduction of total PV+ cells in the cortex, as seen in Fig. 4 f middle panel (current Supplementary Fig. 6d), is a global reduction.

4) Of note, in the supplementary figure 2d the dot plots for total distance and velocity for Cul3+/+ mice looks completely identical. The author should make sure that this is not a mistake in the graph.

Response: We thank the reviewer for the careful examination of our data. With respect to Supplementary Fig. 2d, after having rechecked the raw data of the OF, we can confirm that no mistake has happened in the generation of the graphs. The peculiar identical look of the graphs

comes rather from the fact that the total distance travelled (m) by a mouse and the mean velocity (cm/sec) of its exploration are so tightly linked, that the data point distribution appears identical when plotted.

Reviewer #2 (Remarks to the Author):

In this manuscript, the authors investigate the role of one of the high-risk ASD genes that encode for Cul3 in cortical development. By performing a battery of behavioural test in the constitutive heterozygous Cul3 KO mouse, the authors revealed motor deficits accompanied by aberrant social novelty exploration. Interestingly, deletion of Cul3 in adulthood only impaired rotarod task. The authors found that Cul3 heterozygous mutation results deficits in neural cell migration and cortical lamination as well as changes in spontaneous frequency both of excitatory and inhibitory transmission. No reduction in the dendritic length, branching or in the number of dendrites and spines were revealed. The authors identified up and down regulated proteins in the Cul3^{+/-} and Cul3^{+/fl} Emx1-Cre embryonic cortexes. Among the several proteins, the analysis pointed to proteins involved in the cytoskeletal organization and in particular to PIs3. By increasing Cul3 expression, the authors were able finally to rescue the neural cell migration defects.

The manuscript is interesting and well written, the approaches are state of the art and the quality of the figures is excellent. Several points and limitations are well discussed in the discussion and I think that overall the paper offer important element for possible drug development.

We thank the Reviewer for the positive feedback and the suggestions. We provide a point-by-point response below.

I have some comments:

The rationale underlying the choice of investigate somatosensory cortex is not clear. Indeed, considering the behavioural and anatomical data, I would have expected that the authors would have focus their study on the cerebellum. Do the Emx1-Cre mice also show behavioural deficits?

Response: We studied the *Cul3*^{+/-} brain cyto-architecture and mouse behavior in parallel and, while we did not observe obvious differences in the cytoarchitecture of the cerebellum, we immediately observed the defective layering in the cortex. Since cortical defects are associated with some of the behavioral phenotypes we observe (PMID 28659766, 22099455, 23635870), we decided to focus on the cortex and for simplicity on one region of the cortex. Here we would like to clarify that the cytoarchitecture differences observed in the *Cul3*^{+/-} (as well as *Cul3*^{+/fl} *Emx1-Cre*) are not limited to the somatosensory cortex but spread through the cortex, including e.g. mPFC relevant for social behavior. However, the reviewer is right and Cul3 has probably also a function in the cerebellum, explaining, at least in part, the motor issues observed in our *Cul3*^{+/-} animals. Indeed, the *Cul3*^{+/fl} *Emx1-Cre* mice also show behavioral deficits, such as decreased social interest (PMID: 31455858), however, they lack motor abnormalities, in line with an additional cerebellar role of Cul3. Thus, future studies should aim to analyze the role of Cul3 in the cerebellum.

In the behavioural experiments, it is not clear why the authors only separated between male and female for the motor task and not for the other experiments.

Response: The datasets were split into sexes whenever significant differences between wild-type males and females were observed. We agree however, that the inconsistent presentation of datasets rendered the behavioral description needlessly complicated and therefore now we provide pooled datasets in the main figure and highlighted the minor sex-differences in Supplementary Fig. 2.

The electrophysiological data are not convincing. Indeed, it seems that Cul3 downregulation affects the inhibitory and excitatory spontaneous frequency more than the amplitude. How the E/I ratio was calculated? I would encourage to avoid term as slightly shift or to write “GABAergic transmission is less affected”. Indeed, the comparison has not been done statistically.

Response: We thank the reviewer for this comment. The E/I ratio was calculated as the ratio of sEPSC-to-sIPSC mean amplitudes for each cell. For the revised version of the manuscript we also recorded/analyzed a few additional pyramidal neurons and now we show a clear difference in the amplitude of sEPSC. The sentence “GABA transmission is less affected” was removed.

Although the authors do not find changes in the number of dendrites and spines, they should investigate whether the spine morphology is affected. Along this line, the author should as well investigate the G actin/F actin ratio to have more information regarding the role of actin polymerization.

Response: We thank the reviewer for this suggestion and now include also an analysis of the spine morphology of *Cul3* mutant neurons (Supplementary Fig. 7l), showing no differences between genotypes. Furthermore, we investigated the G-actin/F-actin ratio and could not detect any difference between *Cul3*^{+/-} and wild-type samples (Supplementary Fig. 7k), indicating that the rate of actin polymerization is comparable to wild-type in mutant neurons. This is in line with our *in vitro* observation that the actin-organization but not the amount of filamentous actin is altered in *Cul3*^{+/-} cells, as the same STED imaging parameters could be used for wild-types and mutant cells.

Are Pls3 level normalized by enhancing Cul3 gene transcription?

Response: We thank the reviewer for this comment. We now quantified Pls3 protein levels in CRISPR-rescued *Cul3*^{+/-} NPCs. Indeed, we found that increasing Cul3 levels is sufficient to normalize Pls3 levels in cells (Supplementary Fig. 11b), further supporting the link between Cul3 and Pls3.

Minor:

In Figure 1d, although the data indicate that mutant mice do not reach the same level of motor performance as the control by the third day, male mice show an increase in their performance. I would suggest that the expression, motor learning impairment is too strong and not supported by the data.

Response: We rephrased this statement accordingly.

Reviewer #3 (Remarks to the Author):

This study provides a thorough description of behavioral alterations and cortical malformations arising in an haploinsufficient *Cul3* mouse model. The behavioral analysis is impressive in its rigor and reveals locomotor defects, lack of interest in social novelty and alterations in cognition. This work is also remarkable by its multidisciplinary approach, ranging from the analysis of mouse behavior, histology and physiology to proteomic screening. The manuscript is clearly written, the behavioral analysis is very impressive, but in its present form, the molecular analysis falls short of presenting a clear disease mechanism. Additional experiments attempting to better understand the cause of the cortical malformations and whether those contribute to the behavioral phenotype would dramatically increase the impact of the manuscript. Some key experiments should thus be redesigned to more directly address the claims of the manuscript.

We thank the reviewer for appreciating the multidisciplinary approach and the rigor of our study, as well as for the constructive criticisms. Below, a point-by-point response to her/his suggestions.

Major issues**About the behavioral analysis**

> The social novelty phenotype is sometimes referred in the manuscript to as “reduced social memory” or “interest in social novelty”. In my opinion the latter is a better way to phrase it. Indeed, the behavioral analysis does not convincingly show that lack of interest in social novelty is due to memory alterations since memory retention is not impaired. In addition, interest in social novelty can also be reduced following impairment in the reward circuitry, independently of memory alterations. Additional experiments addressing the implication of memory on the social phenotype could be carried out, for instance using the object recognition test.

Response: We thank the Reviewer for this comment and agree that reduced “interest in social novelty” is the more appropriate description of our observations. Therefore, we rephrased it accordingly in the revised version of the manuscript.

About the reduction of cortical thickness.

>It is unclear how the reduction of cortical thickness occurs in *Cul3* het animals. Migration defects are unlikely to contribute to this phenotype, thus proper analysis of cell cycle progression and cell cycle exit should be performed. Moreover, when looking at embryonic *Cul3* cKO, the thickness reduction is mostly affecting upper layers, pointing to a possible IP defect. This should be addressed and the expression pattern of *Cul3* should also be carefully analyzed in the developing cortex. Reduction of cortical thickness can also arise from change of cell density, this should be looked at.

Response: Unfortunately, we were not able to perform a detailed analysis of *Cul3* protein

expression pattern in the developing cortex, as none of the tested antibodies (Cell Signaling #2759 and Abcam ab194584) provided convincing immunofluorescent staining when compared to the negative control (the cortex of *Cul3^{fl/fl} Emx1-Cre*). However, we investigated *Cul3* expression by making use of single-cell RNA-sequencing data (Mouse Organogenesis Cell Atlas, MOCA), encompassing data from ~2 million cells derived from 61 embryos staged between E9.5 and E13.5 (PMID: 30787437). This data shows moderate *Cul3* expression in a large number of cell types (**Figure A**) in the developing brain (e.g. radial glia, neural progenitors and different types of neurons, highlighted in yellow). Interestingly, no single cell type cluster of the developing brain shows exclusive *Cul3* expression.

Figure A:

Figure A: Single-cell RNA sequencing highlights *Cul3* expression in cell types of the developing brain. The plot was generated via the online tool provided by the study authors: <https://oncoscape.v3.sttrcancer.org/atlas.gs.washington.edu.mouse.rna/genes>

To look at cell proliferation we quantified the number of cells positive for the M-phase marker Phospho-Histone H3 (PHH3) at E14.5. We found similar numbers of PHH3+ cells in the *Cul3^{+/-}* or *Cul3^{fl/fl} Emx1-Cre* cortex, suggesting that cell proliferation is not affected by heterozygous *Cul3* loss (Supplementary Fig. 5d,e, Supplementary Fig. 6a). However, we observed a significant increase in PHH3+ and PHH3+ BrdU+ double labeled cells upon a 2 hours BrdU pulse at E14.5 in the cortex of *Cul3^{fl/fl} Emx1-Cre* embryos (Supplementary Fig. 5d,e), suggesting abnormal cell cycle regulation. A cell cycle defect in radial glia upon homozygous *Cul3* loss, is in line with published data about homozygous *Cul3* deletion. *Cul3*, in mammalian cells, regulates cell cycle via controlling the levels of Cyclin E and p60/katanin. *Cul3^{-/-}* embryos at E7.5 show increased

levels of Cyclin E and a much larger fraction of proliferating cells (PMID: 10500095). *Cul3*^{-/-} cells are characterized by a significant increase in ploidy with abnormally high DNA contents and widespread signs of genetic instability (PMID: 17339333). Homozygous *Cul3* loss further leads to an accumulation of p60/katanin and results in mitotic arrest (PMID: 19261606). These defects in cell cycle regulation result in apoptosis (PMID: 17339333) and are in line with the increased numbers of Cleaved Caspase-3 positive cells in the VZ of our *Cul3*^{fl/fl} *Emx1-Cre* embryos at E16.5 and the corresponding loss of Sox2+ radial glia cells (Supplementary Fig. 5f-i).

Since abnormal cell cycle regulation is observed only in homozygous mutants and cannot explain the thinning of cortical layers in the heterozygous mutant animals (i.e. *Cul3*^{+/-} and *Cul3*^{fl/fl} *Emx1-Cre*) we followed the suggestion of the reviewer and investigated a potential intermediate progenitor phenotype by staining for Tbr2 samples obtained from *Cul3*^{+/-} E14.5 embryos. Yet, we could not observe any differences in the number of intermediate progenitor cells in the SVZ of the heterozygous knockout (Supplementary Fig. 6b).

Lastly, as suggested by the reviewer, we assessed the cell density in cortical layers V/VI. Indeed, we counted a significant increase in the number of cell nuclei in 100x100 μm windows in layers V/VI of *Cul3*^{+/-} animals at P0, indicating that a substantial number of neurons accumulates in those layers (Supplementary Fig. 6c).

About the migration defects.

> Analyzing the position BrdU+ cells in the cortex is not sufficient to take strong conclusion on migration and neurosphere assays are not very useful here, as the identity of migrating cell is unknown (they are also moving too fast as to be bona fide projection neurons). The analysis of the migration defects and their rescue should be done by time-lapse recording on cultured brain slices.

Response: The reviewer is correct, although our *in vitro* experiments provide a good description of the migration defects, time-lapse recording of cultured slices is certainly a gold standard. Therefore, as suggested, we now performed 16-hour time-lapse imaging of acute cortical slices obtained from E17.5 sparsely expressing human *Synapsin-GFP*. In agreement with what observed *in vitro* and with the BrdU labeling experiment we found that neuronal cells in *Cul3*^{+/-} slices move slower and shorter distances compared with wild-type cells. We present these results in Fig. 4 of the revised manuscript.

>The reduction of GABAergic interneurons described in the mutant is not supporting a migration defect, as it could arise from the poor production/specification or even cell survival when Cul3 is not properly expressed

Response: We agree that we lack direct evidence for a migration defect to cause the reduced PV+ cell numbers in the cortex. We rephrased this, nonetheless interesting, observation accordingly in the revised manuscript.

>The *in vitro* data provided with Sir-Actin are interesting but difficult to link to the actual projection neurons defect that harbor a very stereotyped shape during migration, which is not ameboid.

Response: While we agree that the *in vitro* Sir-Actin data does not directly inform on the actin conformation of migrating neurons in mutant animals, it is well established that dynamic changes of the actin-cytoskeleton play a fundamental role for projection neuron migration (reviewed here PMID: 19837469). These changes are needed for the initial polarization of newborn neurons as well as the growth of branched actin networks at the leading edge of the migrating cells. Therefore, while the exact actin dynamic maybe different in neurons, the observed disorganization at the cell front of *Cul3*^{+/-} NPCs *in vitro* might also be relevant for the migration phenotype observed *in vivo* and could be informative for further studies on actin conformation in *Cul3* mutants.

Most importantly, we think that the Sir-Actin data is a relevant connection between Pls3 and the actin cytoskeleton. In order to tighten this point in the revised version of the manuscript we added the analysis of Sir-Actin in Pls3 overexpressing NPCs in a wild-type background. Here, we found that Pls3 overexpression is sufficient to cause a similar actin cytoskeleton conformation as observed in *Cul3*^{+/-} cells (Fig. 7h-m), offering an additional evidence for the tight axes Cul3-Pls3-actin.

About molecular mechanism.

How is reduction of Cul3 leading to increase of Pls3? Is it direct, can the authors provide some data to bridge the gap?

Response: To address this comment we performed an immunoprecipitation (IP) experiment for Pls3. We found that, when pulling down Pls3, we were able to Co-IP Cul3, indicating that Pls3 and Cul3 physically interact (**Figure B**) and suggesting that normally Cul3 tags Pls3 for degradation.

Based on the Co-IP and the qPCR data showing reduced *Pls3* mRNA levels in *Cul3* mutant tissue we are confident that the increase in Pls3 levels is due to a post-transcriptional effect and very likely caused by insufficient degradation (current Fig. 5k). Accordingly, we could show that by increasing *Cul3* mRNA levels through CRISPRa in a *Cul3*^{+/-} background, Pls3 protein levels are normalized, further supporting a tight link between Cul3 and Pls3 proteins (Supplementary Fig. 11b).

Figure B:

Figure B: Co-Immunoprecipitation (Co-IP) indicating an interaction between Pls3 and Cul3. a, Western blot analysis of Pls3-pull-down indicates successful immunoprecipitation (Pls3-IP, arrowheads) and confirms increased levels of the cytoskeletal protein Pls3 in *Cul3*^{+/-} lysates ($n=3$ per genotype). **b,** Interaction between Pls3 and Cul3 was validated by a Co-IP assay confirming Cul3 in the pull-down-samples (Pls3-IP, arrowheads).

Minors

Line 51: Indirectly was meant to be inversely?

Response: We are sorry for the mistake and corrected it in the revised manuscript.

Discuss gender differences in stance length

Response: We now present the pooled data of males and females for the gait characterization, as the sex differences were minor and presenting them separated rendered the behavioral part needlessly complicated.

There are no statistical differences in cerebellar area, thus the sentence 147-148 is not correct

Response: The statement was removed.

Line 212: no demonstration has been given to support that the migration defects is cell intrinsic

Response: The statement was removed.

Line 231: can you explain how the E/I ratio has been obtained?

Response: The E/I ratio was calculated as the ratio of sEPSC-to-sIPSC mean amplitudes for each cell. For the revised version of the manuscript we also recorded/analyzed a few additional pyramidal neurons and now we show a clear difference in the amplitude of sEPSC, in line with what we observed in the E/I ratio.

Reviewer #4 (Remarks to the Author):

Morandell and colleagues investigate the function of the autism-associated Cul3 gene, a ubiquitin ligase, and perform an extensive phenotypic analysis of Cul3 mutant mice. The authors show behavioral abnormalities related to social memory and cognition. They observe cell migration defects in cortex in Cul3 mutants and identify a number of cytoskeleton-associated proteins as potential targets of Cul3 using proteomics. They show that the Pls3 protein is upregulated in Cul3 mutant tissue and that reducing Pls3 levels in Cul3 mutant neural progenitor cells (NPCs) normalizes abnormal migratory behavior of these cells.

The paper uses various Cul3 mouse lines and contains many different types of data, some of which interrupt the flow of the study and have little relevance to the rest of the study (e.g. electrophysiological and morphological analysis in Figure 5); at the same time related data is sometimes spread out over multiple figures (e.g. Figures 3 and 4). Some conclusions are overstated. The design of key experiments (proteomic analysis in Figure 6) that lay the mechanistic basis of the study is very difficult to assess. This makes the paper difficult to read in places. A condensed version focusing on the key finding of the role of Cul3 in cell migration would make this manuscript more accessible.

We thank the reviewer for the positive feedback, the constructive criticisms and the suggestions, below a point-by-point response to the reviewers' comments.

Major Points:

1. Figure 2: the authors test the behavioral effects of heterozygous deletion in late postnatal animals using tamoxifen-injected Cul3^{fl/+} Cag-CreER mice. These animals do not show major behavioral abnormalities, in contrast to constitutive Cul3^{+/-} heterozygous mice (Figure 1), leading the authors to conclude that early developmental function of Cul3 is critical for the appearance of behavioral abnormalities later in life. It should be noted that the lines used in these experiments have a different genetic background. Early-stage Cul3 deletion using tamoxifen-injected Cul3^{fl/+} Cag-CreER mice would have allowed for a better comparison in animals of the same genetic background.

Response: We performed the initial behavioral characterization of the *Cul3* heterozygous

knockout employing *Cul3*^{+/-} animals to mimic the patients *CUL3* haploinsufficiency, as well as to avoid potential secondary effects of a Cre-line. Further, while we agree that using the same genetic background for all the experiments would be ideal, injecting pregnant female mice with large amounts of Tamoxifen to induce Cre-recombination in the offspring is known to have potential severe deleterious effects on the mother and the embryos, especially when injected that early in pregnancy (PMID: 31248325). For example, it is well known that early injection of Tamoxifen often requires delivering the pups by C-section. In addition, side effects from the Tamoxifen on brain development and behavior could have had confounding effects when it comes to the *Cul3*^{+/-} phenotype.

2. Figure 6: the authors perform proteomic analysis on multiple brain regions of different genotypes at various ages using TMT-labelling. Because of the complexity of their experiments, it should be much clearer which conditions have actually been run in the same TMT experiment. The current description in the methods section is unclear and leaves the reader guessing. This is especially the case for the embryonic *Emx1*-cre samples, where n=3/genotype as stated in the Methods does not explain how the experiment is set up within a TMT 11-plex. The design should be clearly outlined in the Methods. A cartoon and description in the figure legend outlining the different conditions of the TMT experiment would help to make the design clear without consulting the Methods.

Response: We are sorry to hear that the experimental plan wasn't fully clear. In the revised manuscript we included a cartoon describing the experimental design of the proteomic analysis, highlighting the conditions that were run in the same TMT experiment (Supplementary Fig. 8a). Additionally, we modified the Method section to improve the description of the experimental design.

3. Figure 6: identification of *Pls3* and *Ina* as potential *Cul3* targets. The authors convincingly show that the levels of *Pls3* and *Ina* are altered in *Cul3* mutant tissue, but there is no direct evidence that these proteins are ubiquitinated by *Cul3*. This should be experimentally demonstrated.

Response: We thank the reviewer for this comment. To answer to this question, we performed a co-immunoprecipitation (Co-IP) experiment for *Pls3*. We found that, when pulling down *Pls3*, we were able to identify *Cul3* in the Co-IP+ sample, indicating that, indeed *Pls3* and *Cul3* interact physically (see above, **Figure B**). Unfortunately, however, we were not able to detect any Ubiquitin signal by Western blot, most likely due to insufficient sensitivity of our method.

Nevertheless, based on the Co-IP and the qPCR data showing reduced *Pls3* mRNA levels in *Cul3* mutant tissue we are confident that the increase in *Pls3* levels is due to a post-transcriptional effect and very likely caused by insufficient degradation (current Fig. 5k).

Accordingly, we could show that by increasing *Cul3* mRNA levels through CRISPRa in a *Cul3*^{+/-} background, Pls3 protein levels are normalized, further supporting a direct link between Cul3 and Pls3 proteins (Supplementary Fig. 11b).

4. Figure 6: the False Discovery Rate of 20% that the authors use to determine up- and down-regulated proteins is very lenient compared to common standards. This is especially obvious in Fig 6 panels c and d where multiple protein groups in the scattered “base” (high ratio/low p-value; meaning high variability) of the volcano plots are considered significant. These would be filtered out when a more stringent FDR is used, as is actually shown by the authors in the supplementary tables S3-5,8 that include a more acceptable 10% FDR. The impression is raised that this leniency is only used so that GO term enrichment analysis can be applied and an overstated link to cytoskeleton organization can be made, while their main targets of interest (Pls3 and INA) are actually very significantly upregulated and would survive harsher filtering. The association between Cul3 mutations and cytoskeletal-associated protein regulation in general is overstated again in the adult dataset where only Vimentin and GFAP are identified just once without any wider cytoskeletal effect. The authors should accept that their datasets do not show major changes in the proteome of their Cul3 mice and not claim overblown conclusions based on too soft filtering. Instead they should keep their focus on the targets that are actually significantly regulated and validate them like Pls3 and INA. In that regard it is interesting that the authors chose to completely ignore Wdfy1, which is reproducibly upregulated in all their datasets but is not a cytoskeletal-associated protein.

Response: We initially chose to use a softer 20% FDR filtering with the aim to provide the full picture of changes (accepting some false positives). Based on the reviewer’s suggestion, however, we now stick to the more stringent 10% FDR filtering in Fig. 5b-d which, indeed, highlights our main protein of interest, Pls3, even more. Accordingly, we have removed the statement regarding cytoskeletal proteins in the adult datasets in the revised manuscript.

Furthermore, we would like to comment on the reviewer’s observation regarding Wdfy1. While it is correct that it is reproducibly upregulated in all our datasets, we decided to not follow up on this hit based on our initial comparison between E16.5 control samples (i.e. *Cul3*^{+/+} vs. *Cul3*^{+/fl}), as Wdfy1 is significantly different between these two control conditions, indicating that its upregulation is likely independent of Cul3 protein levels. We have now included this information in Supplementary Fig. 8a,b.

5. Figure 7: in the text and figure title, the authors imply that the disorganized actin cytoskeleton observed in *Cul3*^{+/-} cells is due to the altered levels of Pls3. To demonstrate this the authors should investigate the actin cytoskeleton in NPCs with altered Pls3 levels (KO and overexpression).

Response: The reviewer is correct. While the study of the actin cytoskeleton in the *Pls3* KO is slightly outside the scope of this work, since not directly linked to the disorder/Cul3, in the revised version of the manuscript we added the analysis of Sir-Actin in *Pls3* overexpressing NPCs in a wild-type background (Fig. 7h-m). Here, we found that *Pls3* overexpression is sufficient to lead to the disorganized actin cytoskeleton observed in *Cul3*^{+/-} cells. We thank the reviewer for this suggestion since this experiment offers an additional evidence of the tight axes Cul3-*Pls3*-actin.

Minor Points:

1. Figure 6e: GO analysis: please clarify the origin of the numbers of proteins (178 up, 140 down) used in this analysis: were these significantly regulated proteins or was this analysis done on all regulated proteins, regardless of whether they are significant or not?

Response: GO analysis was exclusively performed on the *Cul3*^{fl/fl} *Emx1-Cre* dataset, including all significantly up- or down-regulated proteins at an FDR of 20% (proteins listed in Supplementary Table 5), to gain insight into which pathways are perturbed upon *Cul3* loss.

2. Figure 6i: Was Nisch identified as a differentially regulated protein in *Cul3*^{+/-} mice as well? Please clarify.

Response: We apologize for not having mentioned Nisch in the text so far. As highlighted in current Fig. 5d, we detected increased levels of the cytoskeleton interacting Nisch in the *Cul3*^{fl/fl} *Emx1-Cre* proteomics dataset and later validated this hit also by WB, however forgot to mention it in the text (current Fig. 5j). We corrected this mistake in the revised manuscript.

3. Images in 7g-l are difficult to interpret. Please provide clearer examples that show the actin cytoskeleton at the cell's edge.

Response: We thank the reviewer for the comment. For the revised version of the manuscript we now provide clearer examples of the actin organization in our NPCs.

4. The flow of the manuscript could be improved by focusing on the key findings and moving Figure 2, 4f,g, and Figure 5 to the supplement; Figure 3 and 4 could be combined.

Response: We thank the reviewer for this suggestion and, while we believe that Fig. 2 is important for the message of the paper, we now combined Figures 3 and 4 (current Fig. 3) and moved Fig. 5 to the Supplement (current Supplementary Fig. 7). We agree that the flow of the manuscript benefitted from these changes.

Reviewer #1 (Remarks to the Author):

I thank the authors for addressing all my comments thoughtfully and sufficiently. The findings are exciting and I believe will be of great interest to the field.

Reviewer #2 (Remarks to the Author):

The authors replied to all my comments and i am satisfied with the current version of the paper

Camilla Bellone

Reviewer #3 (Remarks to the Author):

The authors nicely reviewed the manuscript. They reply to all my comments and provided new experimental data supporting their claims.

Reviewer #4 (Remarks to the Author):

The authors have convincingly addressed my concerns. Their clarifications and changes have improved the manuscript, which is suitable for publication in my opinion. I'm just not sure about the coIP shown in Figure B of the rebuttal. Here the band of IP'ed Pls3 or Cul3 seems to run higher than the corresponding input band, which seems odd. With the new data demonstrating a tight link between Cul3 and Pls3 levels I don't think this coIP experiment is necessary and could be left out.

2. REVIEWERS' COMMENTS & RESPONSE

Reviewer #1 (Remarks to the Author):

I thank the authors for addressing all my comments thoughtfully and sufficiently. The findings are exciting and I believe will be of great interest to the field.

We are very happy the reviewer found our study's findings exciting, and we share his/her enthusiasm. We are grateful for the reviewer's positive assessment of our revisions.

Reviewer #2 (Remarks to the Author):

The authors replied to all my comments and I am satisfied with the current version of the paper

Camilla Bellone

We thank Dr. Bellone for her helpful suggestions and for endorsing our revised manuscript for publication.

Reviewer #3 (Remarks to the Author):

The authors nicely reviewed the manuscript. They reply to all my comments and provided new experimental data supporting their claims.

We thank this reviewer for the positive evaluation and for her/his helpful comments on our manuscript.

Reviewer #4 (Remarks to the Author):

The authors have convincingly addressed my concerns. Their clarifications and changes have improved the manuscript, which is suitable for publication in my opinion. I'm just not sure about the coIP shown in Figure B of the rebuttal. Here the band of IP'ed Pls3 or Cul3 seems to run higher than the corresponding input band, which seems odd. With the new data demonstrating a tight link between Cul3 and Pls3 levels I don't think this coIP experiment is necessary and could be left out.

We appreciate the reviewer's positive feedback and for recommending our revised manuscript for publication. We agree with the reviewer's observation, the band shift in the Co-IP experiment is peculiar. However, it is conceivable that we enriched the neddylated form of Cul3 when pulling down Pls3. Nedd8 (MW of ~10 kDa) binds to Cul3 when the E3 ubiquitin ligase complex is assembled, explaining the slight shift in molecular weight; in contrast, the non-neddylated protein may be predominant in the native lysates. The cause for the Pls3 bandshift in den co-IP however remains unknown and would demand further investigation. Therefore, we agree with the reviewer to leave the experiment out.